# ON THE EXPRESSIVE POWER OF OVERLAPPING ARCHITECTURES OF DEEP LEARNING

**Or Sharir & Amnon Shashua**
The Hebrew University of Jerusalem
{or.sharir,shashua}@cs.huji.ac.il

## ABSTRACT

Expressive efficiency refers to the relation between two architectures A and B, whereby any function realized by B could be replicated by A, but there exists functions realized by A, which cannot be replicated by B unless its size grows significantly larger. For example, it is known that deep networks are exponentially efficient with respect to shallow networks, in the sense that a shallow network must grow exponentially large in order to approximate the functions represented by a deep network of polynomial size. In this work, we extend the study of expressive efficiency to the attribute of network connectivity and in particular to the effect of "overlaps" in the convolutional process, i.e., when the stride of the convolution is smaller than its filter size (receptive field). To theoretically analyze this aspect of network's design, we focus on a well-established surrogate for ConvNets called *Convolutional Arithmetic Circuits* (ConvACs), and then demonstrate empirically that our results hold for standard ConvNets as well. Specifically, our analysis shows that having overlapping local receptive fields, and more broadly denser connectivity, results in an exponential increase in the expressive capacity of neural networks. Moreover, while denser connectivity can increase the expressive capacity, we show that the most common types of modern architectures already exhibit exponential increase in expressivity, without relying on fully-connected layers.

## 1 INTRODUCTION

One of the most fundamental attributes of deep networks, and the reason for driving its empirical success, is the "Depth Efficiency" result which states that deeper models are exponentially more expressive than shallower models of similar size. Formal studies of Depth Efficiency include the early work on boolean or thresholded circuits (Sipser, 1983; Yao, 1989; Håstad and Goldmann, 1991; Hajnal et al., 1993), and the more recent studies covering the types of networks used in practice (Pascanu et al., 2013; Montúfar et al., 2014; Eldan and Shamir, 2016; Cohen et al., 2016a; Cohen and Shashua, 2016; Telgarsky, 2016; Safran and Shamir, 2016; Raghu et al., 2016; Poole et al., 2016). What makes the Depth Efficiency attribute so desirable, is that it brings exponential increase in expressive power through merely a polynomial change in the model, i.e. the addition of more layers. Nevertheless, depth is merely one among many architectural attributes that define modern networks. The deep networks used in practice consist of architectural features defined by various schemes of connectivity, convolution filter defined by size and stride, pooling geometry and activation functions. Whether or not those relate to expressive efficiency, as depth has proven to be, remains an open question.

In order to study the effect of network design on expressive efficiency we should first define "efficiency" in broader terms. Given two network architectures $A$ and $B$, we say that architecture $A$ is expressively efficient with respect to architecture $B$, if the following two conditions hold: *(i)* any function $\mathbf{h}$ realized by $B$ of size $r_B$ can be realized (or approximated) by $A$ with size $r_A \in \mathcal{O}(r_B)$; *(ii)* there exist a function $\mathbf{h}$ realized by $A$ with size $r_A$, that cannot be realized (or approximated) by $B$, unless $r_B \in \Omega(f(r_A))$ for some super-linear function $f$. The exact definition of the sizes $r_A$ and $r_B$ depends on the measurement we care about, e.g. the number of parameters, or the number of "neurons". The nature of the function $f$ in condition *(ii)* determines the type of efficiency taking place – if $f$ is exponential then architecture $A$ is said to be exponentially efficient with respect to

architecture $B$, and if $f$ is polynomial so is the expressive efficiency. Additionally, we say $A$ is *completely efficient* with respect to $B$, if condition (ii) holds not just for some specific functions (realizable by $A$), but for all functions other than a negligible set.

In this paper we study the efficiency associated with the architectural attribute of convolutions, namely the size of convolutional filters (receptive fields) and more importantly its proportion to their stride. We say that a network architecture is of the *non-overlapping* type when the size of the local receptive field in each layer is equal to the stride. In that case, the sets of pixels participating in the computation of each two neurons in the same layer are completely separated. When the stride is smaller than the receptive field we say that the network architecture is of the *overlapping* type. In the latter case, the *overlapping degree* is determined by the *total* receptive field and stride projected back to the input layer – the implication being that for the overlapping architecture the total receptive field and stride can grow much faster than with the non-overlapping case.

As several studies have shown, non-overlapping convolutional networks do have some theoretical merits. Namely, non-overlapping networks are universal (Cohen et al., 2016a; Cohen and Shashua, 2016), i.e. they can approximate any function given sufficient resources, and in terms of optimization, under some conditions they actually possess better convergence guarantees than overlapping networks. Despite the above, there are only few instances of strictly non-overlapping networks used in practice (e.g. Sharir et al. (2016); van den Oord et al. (2016)), which raises the question of **why are non-overlapping architectures so uncommon?** Additionally, when examining the kinds of architectures typically used in recent years, which employ a mixture of both overlapping and non-overlapping layers, there is a trend of using ever smaller receptive fields, as well as non-overlapping layers having an ever increasing role (Lin et al., 2014; Springenberg et al., 2015; Szegedy et al., 2015). Hence, the most common networks used practice, though not strictly non-overlapping, are increasingly approaching the non-overlapping regime, which raises the question of **why having just slightly overlapping architectures seems sufficient for most tasks?**

In the following sections, we will shed some light on these questions by analyzing the role of overlaps through a surrogate class of convolutional networks called Convolutional Arithmetic Circuits (ConvACs) (Cohen et al., 2016a) – instead of non-linear activations and average/max pooling layers, they employ linear activations and product pooling. ConvACs, as a theoretical framework to study ConvNets, have been the focused of several works, showing, amongst other things, that many of the results proven on this class are typically transferable to standard ConvNets as well (Cohen and Shashua, 2016; 2017). Though prior works on ConvACs have only considered non-overlapping architectures, we suggest a natural extension to the overlapping case that we call Overlapping ConvACs. In our analysis, which builds on the known relation between ConvACs and tensor decompositions, we prove that overlapping architectures are in fact completely and exponentially more efficient than non-overlapping ones, and that their expressive capacity is directly related to their *overlapping degree*. Moreover, we prove that having even a limited amount of overlapping is sufficient for attaining this exponential separation. To further ground our theoretical results, we demonstrate our findings through experiments with standard ConvNets on the CIFAR10 image classification dataset.

## 2 OVERLAPPING CONVOLUTIONAL ARITHMETIC CIRCUITS

In this section, we introduce a class of convolutional networks referred to as Overlapping Convolutional Arithmetic Circuits, or Overlapping ConvACs for short. This class shares the same architectural features as standard ConvNets, including some that have previously been overlooked by similar attempts to model ConvNets through ConvACs, namely, having any number of layers and unrestricted receptive fields and strides, which are crucial for studying overlapping architectures. For simplicity, we will describe this model only for the case of inputs with two spatial dimensions, e.g. color images, and limiting the convolutional filters to the shape of a square.

We begin by presenting a broad definition of a Generalized Convolutional (GC) layer as a fusion of a $1 \times 1$ linear operation with a pooling function – this view of convolutional layers is motivated by the all-convolutional architecture (Springenberg et al., 2015), which replaces all pooling layers with convolutions with stride

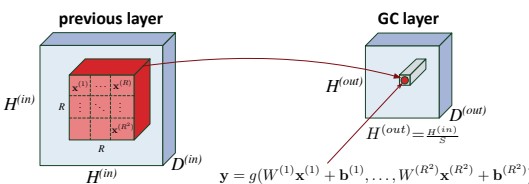

Figure 1: An illustration of a GC Layer.

Figure 2: An illustration of a Generalized Convolutional Network.

greater than 1. The input to a GC layer is a 3-order tensor (multi-dimensional array), having width and height equal to $H^{(\text{in})} \in \mathbb{N}$ and depth $D^{(\text{in})} \in \mathbb{N}$, also referred to as channels, e.g. the input could be a 2D image with RGB color channels. Similarly, the output of the layer has width and height equal to $H^{(\text{out})} \in \mathbb{N}$ and $D^{(\text{out})} \in \mathbb{N}$ channels, where $H^{(\text{out})} = \frac{H^{(\text{in})}}{S}$ for $S \in \mathbb{N}$ that is referred to as the *stride*, and has the role of a sub-sampling operation. Each spatial location $(i, j)$ at the output of the layer corresponds to a 2D window slice of the input tensor of size $R \times R \times D^{(\text{in})}$, extended through all the input channels, whose top-left corner is located exactly at $(i \cdot S, j \cdot S)$, where $R \in \mathbb{N}$ is referred to as its *local receptive field*, or filter size. For simplicity, the parts of window slices extending beyond the boundaries have zero value. Let $\mathbf{y} \in \mathbb{R}^{D^{((out)}}$ be a vector representing the channels at some location of the output, and similarly, let $\mathbf{x}^{(1)}, \ldots, \mathbf{x}^{(R^2)} \in \mathbb{R}^{D^{(\text{in})}}$ be the set of vectors representing the slice, where each vector represents the channels at its respective location inside the $R \times R$ window, then the operation of a GC layer is defined as follows:

$$\mathbf{y} = g(W^{(1)}\mathbf{x}^{(1)} + \mathbf{b}^{(1)}, \ldots, W^{(R^2)}\mathbf{x}^{(R^2)} + \mathbf{b}^{(R^2)}),$$

where $W^{(1)}, \ldots, W^{(R^2)} \in \mathbb{R}^{D^{(out)} \times D^{(in)}}$ and $\mathbf{b}^{(1)}, \ldots, \mathbf{b}^{(R^2)} \in \mathbb{R}^{D^{(out)}}$ are referred to as the weights and biases of the layer, respectively, and $g : \mathbb{R}^{D^{(out)}} \times \cdots \times \mathbb{R}^{D^{(out)}} \to \mathbb{R}^{D^{(out)}}$ is some point-wise pooling function. See fig. 1 for an illustration of the operation a GC layer performs.

With the above definitions, a GC network is simply a sequence of $L$ GC layers, where for $l \in [L] \equiv \{1, \ldots, L\}$, the $l$'th layer is specified by a local receptive field $R^{(l)}$, a stride $S^{(i)}$, $D^{(l)}$ output channels, parameters $\theta^{(l)}$, and a pooling function $g^{(l)}$. For classification tasks, the output of the last layer of the network typically has $1 \times 1$ spatial dimensions, i.e. a vector, where each output channel $y \in [Y] \equiv [D^{(L)}]$ represents the score function of the $y$'th class, denoted by $\mathbf{h}_y$, and inference is perform by $y^* = \arg\max_y h_y(X)$. Oftentimes, it is common to consider the output of the very first layer of a network as a low-level feature representation of the input, which is motivated by the observation that these learned features are typically shared across different tasks and datasets over the same domain (e.g. edge and Gabor filters for natural images). Hence, we treat this layer as a separate fixed "zeroth" convolutional layer referred to as the *representation* layer, where the operation of the layer can be depicted as applying a set of fixed functions $\{f_d : \mathbb{R}^s \to \mathbb{R}\}_{d=1}^M$ to the window slices denoted by $\mathbf{x}_1, \ldots, \mathbf{x}_N \in \mathbb{R}^s$, i.e. the entries of the output tensor of this layer are given by $\{f_d(\mathbf{x}_i)\}_{d \in [M], i \in [N]}$. With these notations, the output of a GC network can be viewed as a function $\mathbf{h}_y(\mathbf{x}_1, \ldots, \mathbf{x}_N)$. The entire GC network is illustrated in fig. 2.

Given a non-linear point-wise activation function $\sigma(\cdot)$ (e.g. ReLU), then setting all pooling functions to average pooling followed by the activation, i.e. $g(\mathbf{x}^{(1)}, \ldots, \mathbf{x}^{(R^2)})_c = \sigma\left(\sum_{i=1}^{R^2} x_c^{(i)}\right)$ for $c \in [D^{(\text{out})}]$, give rise to the common all-convolutional network with $\sigma(\cdot)$ activations, which served as the initial motivation for our formulation. Alternatively, choosing instead a product pooling function, i.e. $g(\mathbf{x}^{(1)}, \ldots, \mathbf{x}^{(R^2)})_c = \prod_{i=1}^{R^2} x_c^{(i)}$ for $c \in [D^{(\text{out})}]$, results in an Arithmetic Circuit, i.e. a circuit containing just product and sum operations, hence it is referred to as a Convolutional Arithmetic Circuit, or ConvAC. It is important to emphasize that ConvACs, as originally introduced by Cohen et al. (2016a), are typically described in a very different manner, through the language of tensor decompositions (see app. A for background). Since vanilla ConvACs can be seen as an alternating sequence of $1 \times 1$ convolutions and non-overlapping product pooling layers, then the two formulations coincide when all GC layers are non-overlapping, i.e. for all $l \in [L]$, $R^{(l)} = S^{(l)}$. If, however, some of the layers are overlapping, i.e. there exists $l \in [L]$ such that $R^{(l)} > S^{(l)}$, then our formulation through GC layers diverges, and give rise to what we call *Overlapping ConvACs*.

Given that our model is an extension of the ConvACs framework, it inherits many of its desirable attributes. First, it shares most of the same traits as modern ConvNets, i.e. locality, sharing and pooling. Second, it can be shown to form a universal hypotheses space (Cohen et al., 2016a). Third, its underlying operations lend themselves to mathematical analysis based on measure theory and tensor analysis (Cohen et al., 2016a). Forth, through the concept of generalized tensor decompositions (Cohen and Shashua, 2016), many of the theoretical results proven on ConvACs could be transferred to standard ConvNets with ReLU activations. Finally, from an empirical perspective, they tend to work well in many practical settings, e.g. for optimal classification with missing data (Sharir et al., 2016), and for compressed networks (Cohen et al., 2016b).

While we have just established that the non-overlapping GC Network with a product pooling function is equivalent to vanilla ConvACs, one might wonder if using overlapping layers instead could diminish what these overlapping networks can represent. We show that not only is it not the case, but prove the more general claim that a network of a given architecture can realize exactly the same functions as networks using smaller local receptive fields, which includes the non-overlapping case.

**Proposition 1.** *Let A and B be two GC Networks with a product pooling function. If the architecture of B can be derived from A through the removal of layers with $1\times1$ stride, or by decreasing the local receptive field of some of its layers, then for any choice of parameters for B, there exists a matching set of parameters for A, such that the function realized by B is exactly equivalent to A. Specifically, A can realize any non-overlapping network with the same order of strides (excluding $1\times1$ strides).*

*Proof sketch.* This follows from two simple claims: (i) a GC layer can produce an output equivalent to that of a GC layer with a smaller local receptive field, by "zeroing" its weights beyond the smaller local receptive field; and (ii) GC layers with $1\times1$ receptive fields can be set such that their output is equal to their input, i.e. realize the identity function. With these claims, the local receptive fields of A can be effectively shrank to match the local receptive fields of B, and any additional layers of A with stride $1\times1$ could be set such that they are realizing the identity mapping, effectively "removing" them from A. See app. C.2 for a complete proof. □

Proposition 1 essentially means that overlapping architectures are just as expressive as non-overlapping ones of similar structure, i.e. same order of non-unit strides. As we recall, this satisfies the first condition of the efficiency property introduced in sec. 1, and does so regardless if we measure the size of a network as the number of parameters, or the number of "neurons"[1]. In the following section we will cover the preliminaries required to show that overlapping networks actually lead to an increase in expressive capacity, which under some settings results in an exponential gain, proving that the second condition of expressive efficiency holds as well.

## 3 ANALYZING EXPRESSIVE EFFICIENCY THROUGH GRID TENSORS

In this section we describe our methods for analyzing the expressive efficiency of overlapping ConvACs that lay the foundation for stating our theorems. A minimal background on tensor analysis required to follow our work can be found in sec. 3.1, followed by presenting our methods in sec. 3.2.

### 3.1 PRELIMINARIES

In this sub-section we cover the minimal background on tensors analysis required to understand our analysis. A tensor $\mathcal{A} \in \mathbb{R}^{M_1 \otimes \cdots \otimes M_N}$ of order $N$ and dimension $M_i$ in each mode $i \in [N] \equiv \{1, \ldots, N\}$, is a multi-dimensional array with entries $\mathcal{A}_{d_1,\ldots,d_N} \in \mathbb{R}$ for all $i \in [N]$ and $d_i \in [M_i]$. For simplicity, henceforth we assume that all dimensions are equal, i.e. $M \equiv M_1 = \ldots = M_N$. One of the central concepts in tensor analysis is that of *tensor matricization*, i.e. rearranging its entries to the shape of a matrix. Let $P \cup Q = [N]$ be a disjoint partition of its indices, such that $P = \{p_1, \ldots, p_{|P|}\}$ with $p_1 < \ldots < p_{|P|}$, and $Q = \{q_1, \ldots, q_{|Q|}\}$ with $q_1 < \ldots < q_{|Q|}$. The matricization of $\mathcal{A}$ with respect to the partition $P \cup Q$, denoted by $[\![\mathcal{A}]\!]_{P,Q}$, is the $M^{|P|}$-by-$M^{|Q|}$ matrix holding the entries of $\mathcal{A}$, such that for all $i \in [N]$ and $d_i \in [M]$ the entry $A_{d_1,\ldots,d_N}$ is placed

---

[1]We take here the broader definition of a "neuron", as any one of the scalar values comprising the output array of an arbitrary layer in a network. In the case the output array is of width and height equal to $H$ and $C$ channels, then the number of such "neurons" for that layer is $H^2 \cdot C$.

in row index $1 + \sum_{t=1}^{|P|}(d_{p_t} - 1)M^{|P|-t}$ and column index $1 + \sum_{t=1}^{|Q|}(d_{q_t} - 1)M^{|Q|-t}$. Lastly, the tensors we study in this article originate by examining the values of some given function at a set of predefined points and arranging them in a tensor referred to as the *grid tensor* of the function. Formally, let $f : \mathbb{R}^s \times \ldots \times \mathbb{R}^s \to \mathbb{R}$ be a function, and let $\{\mathbf{x}^{(1)}, \ldots, \mathbf{x}^{(M)} \in \mathbb{R}^s\}$ be a set of vectors called *template vectors*, then the grid tensor of $f$ is denoted by $\mathcal{A}(f) \in \mathbb{R}^{M \otimes \ldots \otimes M}$ and defined by $\mathcal{A}(f)_{d_1,\ldots,d_N} = f(\mathbf{x}^{(d_1)}, \ldots, \mathbf{x}^{(d_N)})$ for all $d_1, \ldots, d_N \in [M]$.

## 3.2 BOUNDING THE SIZE OF NETWORKS VIA GRID TENSORS

We begin with a discussion on how to have a well-defined measure of efficiency. We wish to compare the efficiency of non-overlapping ConvACs to overlapping ConvACs, for a fixed set of $M$ representation functions (see sec. 2 for definitions). While all functions realizable by non-overlapping ConvACs with shared representation functions lay in the same function subspace (see Cohen et al. (2016a)), this is not the case for overlapping ConvACs, which can realize additional functions outside the sub-space induced by non-overlapping ConvACs. We cannot therefore compare both architectures directly, and need to compare them through an auxiliary objective. Following the work of Cohen and Shashua (2016), we instead compare architectures through the concept of grid tensors, and specifically, the grid tensor defined by the output of a ConvAC, i.e. the tensor $\mathcal{A}(\mathbf{h})$ for $\mathbf{h}(\mathbf{x}_1, \ldots, \mathbf{x}_N)$. Unlike with the ill-defined nature of directly comparing the functions of realized by ConvACs, Cohen and Shashua (2016) proved that assuming the fixed representation functions are linearly independent, then there exists template vectors $\mathbf{x}^{(1)}, \ldots, \mathbf{x}^{(M)}$, for which any non-overlapping ConvAC architecture could represent all possible grid tensors over these templates, given sufficient number of channels at each layer. More specifically, if $F_{ij} = f_i(\mathbf{x}^{(j)})$, then these template vector are chosen such that $F$ is non-singular. Thus, once we fix a set of linearly independent representation functions, we can compare different ConvACs, whether overlapping or not, on the minimal size required for them to induce the same grid tensor, while knowing such a finite number always exists.

One straightforward direction for separating between the expressive efficiency of two network architectures A and B is by examining the ranks of their respective matricized grid tensors. Specifically, Let $\mathcal{A}(\mathbf{h}^{(A)})$ and $\mathcal{A}(\mathbf{h}^{(B)})$ denote the grid tensors of A and B, respectively, and let $(P, Q)$ be a partition of $[N]$, then we wish to find an upper-bound on the rank of $[\![\mathcal{A}(\mathbf{h}^{(A)})]\!]_{P,Q}$ as a function of its size on one hand, while showing on the other hand that $\text{rank}\left([\![\mathcal{A}(\mathbf{h}^{(B)})]\!]_{P,Q}\right)$ can be significantly greater. One benefit of studying efficiency through a matrix rank is that not only we attain separation bounds for exact realization, but also immediately gain access to approximation bounds by examining the singular values of the matricized grid tensors. This brings us to the following lemma, which connects upper-bounds that were previously found for non-overlapping ConvACs (Cohen and Shashua, 2017), with the grid tensors induced by them (see app. C.1 for proof):

**Lemma 1.** *Let $h_y(\mathbf{x}_1, \ldots, \mathbf{x}_N)$ be a score function of a non-overlapping ConvAC with a fixed set of $M$ linearly independent and continuous representation functions, and $L$ GC layers. Let $(P, Q)$ be a partition dividing the spatial dimensions of the output of the representation layer into two equal parts, either along the horizontal or vertical axis, referred to as the "left-right" and "top-bottom" partitions, respectively. Then, for any template vectors such that $F$ is non-singular and for any choice of the parameters of the network, it holds that $\text{rank}\left([\![\mathcal{A}(\mathbf{h}_y)]\!]_{P,Q}\right) \leq D^{(L-1)}$.*

Lemma 1 essentially means that it is sufficient to show that overlapping ConvACs can attain ranks super-polynomial in their size to prove they are exponentially efficient with respect to non-overlapping ConvACs. In the next section we analyze how the overlapping degree is related to the rank, and under what cases it leads to an exponentially large rank.

## 4 THE EXPRESSIVE EFFICIENCY OF OVERLAPPING ARCHITECTURES

In this section we analyze the expressive efficiency of overlapping architectures. We begin by defining our measures of the overlapping degree that will used in our claims, followed by presenting our main results in sec. 4.2. For the sake of brevity, an additional set of results, in light of the recent work by Cohen and Shashua (2017) on "Pooling Geometry", is deferred to app. B.

## 4.1 THE OVERLAPPING DEGREE OF A NETWORK

To analyze the efficiency of overlapping archi-
tectures, we will first formulate more rigorously
the measurement of the overlapping degree of
a given architecture. As mentioned in sec. 1,
we do so by defining the concepts of the *to-
tal receptive field* and *total stride* of a given
layer $l \in [L]$, denoted by $T_R^{(l)}$ and $T_S^{(l)}$, re-
spectively. Both measurements could simply be
thought of as projecting the accumulated local
receptive fields (or strides) to the the first layer,
as illustrated in fig. 3, which represent a type of
global statistics of the architecture. However,
note that proposition 1 entails that a given ar-
chitecture could have a smaller *effective* total
receptive field, for some settings of its parame-
ters. This leads us to define the $\alpha$-minimal total

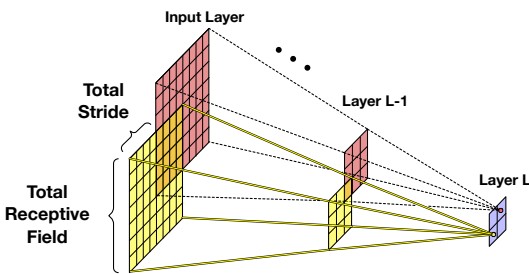

Figure 3: Illustrating the total receptive field and
total stride attributes for the $L$'th layer, which
could be seen as the projected receptive field and
stride with respect to the input layer. Together,
they capture the overlapping degree of a network.

receptive field, for any $\alpha \in \mathbb{R}_+$, as the smallest effective total receptive field still larger than $\alpha$,
which we denote by $T_R^{(l,\alpha)}$. The exact definitions of the above concepts are formulated as follows:

$$T_S^{(l)} \equiv T_S^{(l)}(S^{(1)}, \ldots, S^{(l)}) \equiv \begin{cases} \prod_{i=1}^{l} S^{(i)} & l \geq 1 \\ 1 & l = 0 \end{cases} \tag{1}$$

$$T_R^{(l)} \equiv T_R^{(l)}(R^{(1)}, S^{(1)}, \ldots, R^{(l)}, S^{(l)}) \equiv R^{(l)} \cdot T_S^{(l-1)} + \sum_{k=1}^{l-1} \left( R^{(k)} - S^{(k)} \right) \cdot T_S^{(k-1)} \tag{2}$$

$$T_R^{(l,\alpha)} \equiv T_R^{(l,\alpha)}(R^{(1)}, S^{(1)}, \ldots, R^{(l)}, S^{(l)}) \equiv \underset{\substack{\forall i \in [l], S^{(i)} \leq t_i \leq R^{(i)} \\ T_R^{(l)}(t_1, S^{(1)}, \ldots, t_l, S^{(l)}) > \alpha}}{\operatorname{argmin}} T_R^{(l)}(t_1, S^{(1)}, \ldots, t_l, S^{(l)}) \tag{3}$$

where we omitted the arguments of $T_S^{(l-1)}$ and $T_S^{(k-1)}$ for the sake of visual compactness.

Notice that for non-overlapping networks the total receptive field always equals the total stride, and
that only at the end of the network, after the spatial dimension collapses to $1 \times 1$, does the the total
receptive field grow to encompass the entire size of the representation layer. For overlapping net-
works this is not the case, and the total receptive field could grow much faster. Intuitively, this means
that values in regions of the input layer that are far apart would be combined by non-overlapping
networks only near the last layers of such networks, and thus non-overlapping networks are effec-
tively shallow in comparison to overlapping networks. Base on this intuition, in the next section we
analyze networks with respect to the point at which their total receptive field is large enough.

## 4.2 MAIN RESULTS

With all the preliminaries in place, we are ready to present our main result:

**Theorem 1.** *Assume a ConvAC with a fixed representation layer having $M$ output channels and both
width and height equal to $H$, followed by $L$ GC layers, where the $l$'th layer has a local receptive
field $R^{(l)}$, a stride $S^{(l)}$, and $D^{(l)}$ output channels. Let $K \in [L]$ be a layer with a total receptive
field $T_R^{(K)} \equiv T_R^{(K)}(R^{(1)}, S^{(1)}, \ldots, R^{(K)}, S^{(K)})$, such that $T_R^{(K)} > \frac{H}{2}$. Then, for any choice of
parameters, except a null set (with respect to the Lebesgue measure), and for any template vectors
such that $F$ is non-singular, the following equality holds:*

$$\operatorname{rank}\left(\left[\!\left[\mathcal{A}(\mathbf{h}_y)\right]\!\right]_{P,Q}\right) \geq D^{\left\lfloor \frac{H - T_R^{(K, \lfloor H/2 \rfloor)}}{T_S^{(K)}} + 1 \right\rfloor \cdot \left\lceil \frac{H}{T_S^{(K)}} \right\rceil} \tag{4}$$

*where $(P, Q)$ is either the "left-right" or the "top-bottom" partitions and
$D \equiv \min\{M, D^{(K)}, \frac{1}{2} \min_{1 \leq l \leq K} D^{(l)}\}$.*

*Proof sketch.* Because the entries of the matricized grid tensors are polynomials in the parameters,
then according to a lemma by Sharir et al. (2016), if there is a single example that attains the above

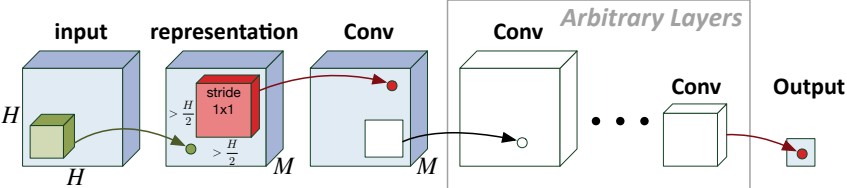

Figure 4: A network architectures beginning with large local receptive fields greater than $N/2$ and at least $M$ output channels. According to theorem 1, for almost all choice of parameters we obtain a function that cannot be approximated by a non-overlapping architecture, if the number of channels in its next to last layer is less than $M^{\frac{H^2}{2}}$.

lower-bound on the rank, then it occurs almost everywhere with respect to the Lebesgue measure on the Euclidean space of the parameters.

Given the last remark, the central part of our proof is simply the construction of such an example. First we find a set of parameters for the simpler case where the first GC layer is greater than a quarter of the input, satisfying the conditions of the theorem. The motivation behind the specific construction is the pairing of indices from each side of the partition, such that they are both in the same local receptive field, and designing the filters such that the output of each local application of them defines a mostly diagonal matrix of rank $D$, with respect to these two indices. The rest of the parameters are chosen such that the output of the entire network results in a product of the entries of these matrices. Under matricization, this results in a matrix who is equivalent[2] to a Kronecker product of mostly diagonal matrices. Thus, the matricization rank is equal to the product of the ranks of these matrices, which results in the exponential form of eq. 4. Finally, we extend the above example to the general case, by realizing the operation of the first layer of the above example through multiple layers with small local receptive fields. See app. C.1 for the definitions and lemmas we rely on, and see app. C.3 for a complete proof. □

Combined with Lemma 1, it results in the following corollary:

**Corollary 1.** *Under the same setting as theorem 1, and for all choices of parameters of an overlapping ConvAC, except a negligible set, any non-overlapping ConvAC that realizes (or approximates) the same grid tensor must be of size at least:*

$$D^{\left\lfloor \frac{H - T_R^{(K,\lfloor H/2 \rfloor)}}{T_S^{(K)}} + 1 \right\rfloor \cdot \left\lceil \frac{H}{T_S^{(K)}} \right\rceil}.$$

While the complexity of the generic lower-bound above might seem incomprehensible at first, its generality gives us the tools to analyze practically any kind of feed-forward architecture. As an example, we can analyze the lower bound for the well known GoogLeNet architecture (Szegedy et al., 2015), for which the lower bound equals $32^{98}$, making it clear that using a non-overlapping architecture for this case is infeasible. Next, we will focus on specific cases for which we can derive more intelligible lower bounds.

According to theorem 1, the lower bound depends on the first layer for which its total receptive field is greater than a quarter of the input. As mentioned in the previous section, for non-overlapping networks this only happens after the spatial dimension collapses to $1 \times 1$, which entails that both the total receptive field and total stride would be equal to the width $H$ of the representation layer, and substituting this values in eq. 4 results simply in $D$ – trivially meaning that to realize one non-overlapping network by another non-overlapping network, the next to last layer must have at least half the channels of the target network.

On the other extreme, we can examine the case where the first GC layer has a local receptive field $R$ greater than a quarter of its input, i.e. $R > H/2$. Since the layers following the first GC layer do not affect the lower bound in this case, it applies to any arbitrary sequence of layers as illustrated in fig. 4. For simplicity we will also assume that the stride $S$ is less than $H/2$, and that $\frac{H}{2}$ is evenly divided

---

[2] Two matrices are equivalent if one could be converted to the other by elementary row or column operations.

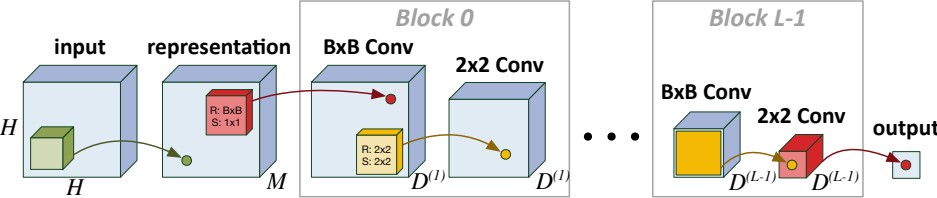

Figure 5: The common network architecture of alternating $B\times B$ "conv" and $2\times 2$ "pooling" layers. If $B \leq {}^H/5+1$ and $D^{(l)} \geq 2M$ for all $1 \leq l < L$, then the lower bound of theorem 1 for this network results in $M^{\frac{(2B-1)^2}{4}}$.

by $S$. In this case the $\frac{H}{2}$-minimal receptive field equals to $\frac{H}{2} + 1$, and thus the lower bound results in $D^{\frac{H^2}{2S}}$. Consider the case of $D = M$ and $S = 1$, then a non-overlapping architecture that satisfies this lower bound is of the order of magnitude at which it could already represent any possible grid tensor. This demonstrate our point from the introduction, that through a a polynomial change in the architecture, i.e. increasing the receptive field, we get an exponential increase in expressivity.

Though the last example already demonstrates that a polynomially sized overlapping architecture could lead to an exponential separation, in practice, employing such large convolutions is very resource intensive. The common best practice is to use multiple small local receptive fields of size $B \times B$, where the typical values are $B = 3$ or $B = 5$, separated by a $2 \times 2$ "pooling" layers, i.e. layers with both stride and local receptive field equal to $2 \times 2$. For simplicity, we assume that $H = 2^L$ for some $L \in \mathbb{N}$. See fig. 5 for an illustration of such a network. Analyzing the above network with theorem 1 results in the following proposition:

**Proposition 2.** *Consider a network comprising a sequence of GC blocks, each block begins with a layer whose local receptive field is $B\times B$ and its stride $1\times 1$, followed by a layer with local receptive field $2\times 2$ and stride $2\times 2$, where the output channels of all layers are at least $2M$, and the spatial dimension of the representation layer is $H\times H$ for $H=2^L$. Then, the lower bound describe by eq. 4 for the above network is greater than or equal to:*

$$\tau(B,H) \equiv M^{\frac{(2B-1)^2}{2}\cdot\left(1+\frac{2B-2}{H}\right)^{-2}} = M^{\frac{H^2}{2}\cdot\left(1+\frac{H-1}{2B-1}\right)^{-2}},$$

*whose limits are $\lim_{B\to\infty} \tau(B,H) = M^{\frac{H^2}{2}}$ and $\lim_{H\to\infty} \tau(B,H) = M^{\frac{(2B-1)^2}{2}}$. Finally, assuming $B \leq \frac{H}{5} + 1$, then $\tau(B,H) \geq M^{\frac{(2B-1)^2}{4}}$.*

*Proof sketch.* We first find a closed-form expression for the total receptive field and stride of each of the $B\times B$ layers in the given network. We then show that for layers whose total receptive field is greater than $\frac{H}{2}$, its $\alpha$-minimal total receptive field, for $\alpha=\frac{H}{2}$, is equal to $\frac{H}{2}+1$. We then use the above to find the first layer who satisfies the conditions of theorem 1, and then use our closed-forms expressions to simplify the general lower bound for this case. See app. C.4 for a complete proof. ☐

In particular, for the typical values of $M = 64$, $B = 5$, and $H \geq 20$, the lower bound is at least $64^{20}$, which demonstrates that even having a small amount of overlapping already leads to an exponential separation from the non-overlapping case. When $B$ grows in size, this bound approaches the earlier result we have shown for large local receptive fields encompassing more than a quarter of the image. When $H$ grows in size, the lower bound is dominated strictly by the local receptive fields. Also notice that based on proposition 2, we could also derive a respective lower bound for a network following VGG style architecture (Simonyan and Zisserman, 2014), where instead of a single convolutional layer before every "pooling" layer, we have $K$ layers, each with a local receptive field of $C \times C$. Under this case, it is trivial to show that the bound from proposition 2 holds for $B = K \cdot (C - 1) + 1$, and under the typical values of $C = 3$ and $K = 2$ it once again results in a lower bound of at least $64^{20}$.

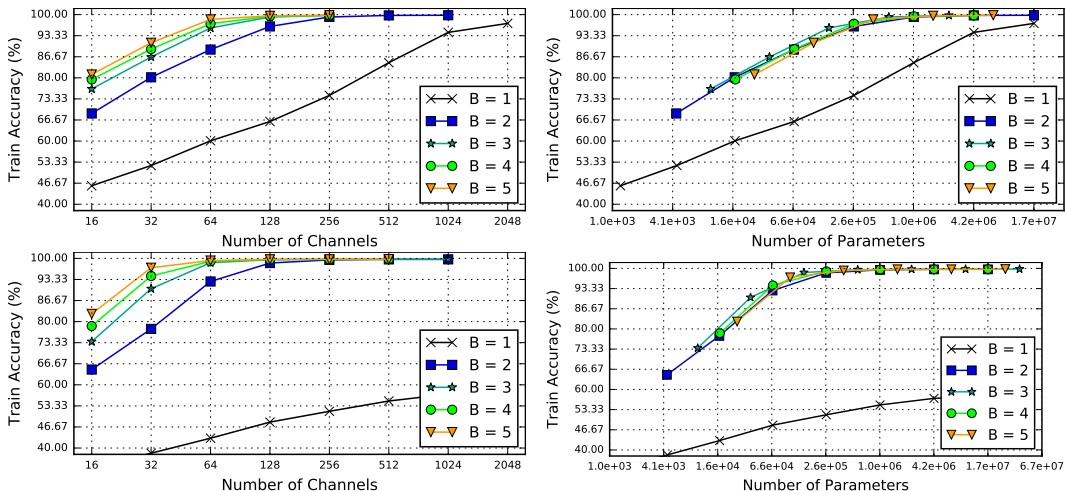

Figure 6: Training accuracies of standard ConvNets on CIFAR-10 with data augmentations, where the results of spatial augmentations presented at the top row, and color augmentations at the bottom row. Each network follows the architecture of proposition 2, with with receptive field $B$ and using the same number of channels across all layers, as specified by the horizontal axis of left plot. We plot the same results with respect to the total number of parameters in the right plot.

## 5 EXPERIMENTS

In this section we show that the theoretical results of sec. 4.2 indeed hold in practice. In other words, there exists tasks that require the highly expressive power of overlapping architectures, on which non-overlapping architectures would have to grow by an exponential factor to achieve the same level of performance. We demonstrate this phenomenon on standard ConvNets with ReLU activations that follow the same architecture that was outlined in proposition 2, while varying the number of channels and the size of the receptive field of the $B{\times}B$ "conv" layers. The only change we made, was to replace the $2{\times}2$-"pooling" layers of the convolutional type, with the standard $2{\times}2$-max-pooling layers, and using the same number of channels across all layers. This was done for the purpose of having all the learned parameters located only at the (possibly) overlapping layers. More specifically, the network has 5 blocks, each starting with a $B{\times}B$ convolution with $C$ channels, stride $1{\times}1$, and ReLU activation, and then followed by $2{\times}2$ max-pooling layer. After the fifth "conv-pool", there is a final dense layer with 10 outputs and softmax activations.

We train each of these networks for classification over the CIFAR-10 dataset, with two types of data augmentation schemes: (i) spatial augmentations, i.e. randomly translating (up to 3 pixels in each direction) and horizontally flipping each image, and (ii) color augmentations following Dosovitskiy et al. (2014), i.e. randomly adding a constant shift (at most $\pm0.3$) to the hue, saturation, and luminance, for each attribute separately, and in addition randomly sampling a multiplier (in the range $[0.5, 1.5]$) just to the saturation and luminance. Though typically data augmentation is only used for the purpose of regularization, we employ it for the sole purpose of raising the hardness of the regular CIFAR-10 dataset, as even small networks can already overfit and effectively memorize its small dataset. We separately test both the spatial and color augmentation schemes to emphasize that our empirical results cannot be explained simply by spatial-invariance type arguments. Finally, the training itself is carried out for 300 epochs with ADAM (Kingma and Ba, 2015) using its standard hyper-parameters, at which point the loss of the considered networks have stopped decreasing. We report the training accuracy over the augmented dataset in fig. 6, where for each value of the receptive field $B$, we plot its respective training accuracies for variable number of channels $C$. The source code for reproducing the above experiments and plots can be found at https://github.com/HUJI-Deep/OverlapsAndExpressiveness.

It is quite apparent that the greater $B$ is chosen, the less channels are required to achieve the same accuracy. Moreover, for the non-overlapping case of $B{=}1$, more than 2048 channels are required to reach the same performance of networks with $B{>}2$ and just 64 channels under the spatial aug-

mentations – which means effectively exponentially more channels were required. Even more so, under the color augmentations, we were not able to train non-overlapping networks to reach even the smallest overlapping network ($B = 2$ and $C = 16$). In terms of total number of parameters, there is a clear separation between the overlapping and the non-overlapping types, and we once again see more than an order of magnitude increase in the number of parameters between an overlapping and non-overlapping architectures that achieve similar training accuracy. As a somewhat surprising result, though based only on our limited experiments, it appears that for the same number of parameters, all overlapping networks attain about the same training accuracy, suggesting perhaps that having the smallest amount of overlapping already attain all the benefits overlapping provides, and that increasing it further does not affect the performance in terms of expressivity.

As final remark, we also wish to acknowledge the limitations of drawing conclusions strictly from empirical experiments, as there could be alternative explanations to these observations, e.g. the effects overlapping has on the optimization process. Nevertheless, our theoretical results suggests this is less likely the case.

## 6 DISCUSSION

The common belief amongst deep learning researchers has been that depth is one of the key factors in the success of deep networks – a belief formalized through the depth efficiency conjecture. Nevertheless, depth is one of many attributes specifying the architecture of deep networks, and each could potentially be just as important. In this paper, we studied the effect overlapping receptive fields have on the expressivity of the network, and found that having them, and more broadly denser connectivity, results in an exponential gain in the expressivity that is orthogonal to the depth.

Our analysis sheds light on many trends and practices in contemporary design of neural networks. Previous studies have shown that non-overlapping architectures are already universal (Cohen et al., 2016a), and even have certain advantages in terms of optimization (Brutzkus and Globerson, 2017), and yet, real-world usage of non-overlapping networks is scarce. Though there could be multiple factors involved, our results clearly suggest that the main culprit is that non-overlapping networks are significantly handicapped in terms of expressivity compared to overlapping ones, explaining why the former are so rarely used. Additionally, when examining the networks that are commonly used in practice, where the majority of the layers are of the convolutional type with very small receptive field, and only few if any fully-connected layers (Simonyan and Zisserman, 2014; Springenberg et al., 2015; He et al., 2016), we find that though they are obviously overlapping, their overlapping degree is rather low. We showed that while denser connectivity can increase the expressive capacity, even in the most common types of modern architectures already exhibit exponential increase in expressivity, without relying on fully-connected layers. This could partly explain that somewhat surprising observation, as it is probable that such networks are sufficiently expressive for most practical needs simply because they are already in the exponential regime of expressivity. Indeed, our experiments seems to suggests the same, in which we saw that further increases in the overlapping degree beyond the most limited overlapping case seems to have insignificant effects on performance – a conjecture not quite proven by our current work, but one we wish to investigate in the future.

There are relatively few other works which have studied the role of receptive fields in neural networks. Several empirical works (Li and Perona, 2005; Coates et al., 2011; Krizhevsky et al., 2012) have demonstrated similar behavior, showing that the classification accuracy of networks can sharply decline as the degree of overlaps is decreased, while also showing that gains from using very large local receptive fields are insignificant compared to the increase in computational resources. Other works studying the receptive fields of neural networks have mainly focused on how to learn them from the data (Coates and Ng, 2011; Jia et al., 2012). While our analysis has no direct implications to those specific works, it does lay the ground work for potentially guiding architecture design, through quantifying the expressivity of any given architecture. Lastly, Luo et al. (2016) studied the *effective total receptive field* of different layers, a property of a similar nature to our total receptive field, where they measure the the degree to which each input pixel is affecting the output of each activation. They show that under common random initialization of the weights, the effective total receptive field has a gaussian shape and is much smaller than the maximal total receptive field. They additionally demonstrate that during training the effective total receptive field grows in size, and suggests that weights should be initialized such that the initial effective receptive field is

large. Their results strengthen our theory, by showing that trained networks tend to maximize their effective receptive field, taking full potential of their expressive capacity.

To conclude, we have shown both theoretically and empirically that overlapping architectures have an expressive advantage compared to non-overlapping ones. Our theoretical analysis is grounded on the framework of ConvACs, which we extend to overlapping configurations. Though are proofs are limited to this specific case, previous studies (Cohen and Shashua, 2016) have already shown that such results could be transferred to standard ConvNets as well, using most of the same mathematical machinery. While adapting our analysis accordingly is left for future work, our experiments on standard ConvNets (see sec. 5) already suggest that the core of our results should hold in this case as well. Finally, an interesting outcome of moving from non-overlapping architectures to overlapping ones is that the depth of a network is no longer capped at $\log_2 (\textit{input size})$, as has been the case in the models investigated by Cohen et al. (2016a) – a property we will examine in future works

### ACKNOWLEDGMENTS

This work is supported by Intel grant ICRI-CI #9-2012-6133, by ISF Center grant 1790/12 and by the European Research Council (TheoryDL project).

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

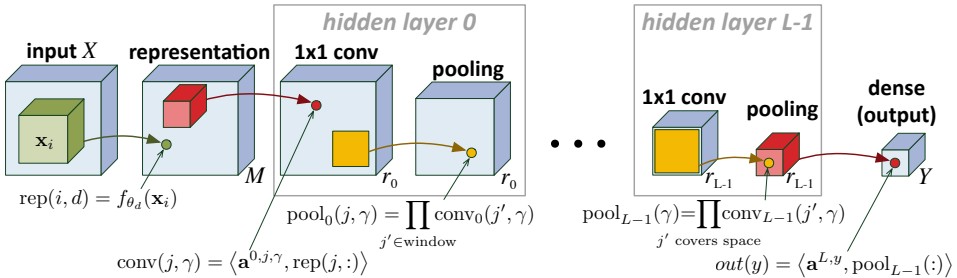

Figure 7: The original Convolutional Arithmetic Circuits as presented by Cohen et al. (2016a).

## A BACKGROUND ON CONVOLUTIONAL ARITHMETIC CIRCUITS

We base our analysis on the convolutional arithmetic circuit (ConvAC) architecture introduced by Cohen et al. (2016a), which is illustrated by fig. 7, and can be simply thought of as a regular ConvNet, but with linear activations and product pooling layers, instead of the more common non-linear activations (e.g. ReLU) and average/max pooling. More specifically, each point in the input space of the network, denoted by $X = (\mathbf{x}_1, \ldots, \mathbf{x}_N)$, is represented as an $N$-length sequence of $s$-dimensional vectors $\mathbf{x}_1, \ldots, \mathbf{x}_N \in \mathbb{R}^s$. $X$ is typically thought of as an image, where each $\mathbf{x}_i$ corresponds to a local patches from that image. The first layer of the network is referred to as the representation layer, consisting of applying $M$ representation functions $f_{\theta_1}, \ldots, f_{\theta_M} : \mathbb{R}^s \to \mathbb{R}$ on each local patch $\mathbf{x}_i$, giving rise to $M$ feature maps. Under the common setting, where the representation functions are selected to be $f_{\theta_d}(\mathbf{x}) = \sigma(\mathbf{w}_d^T \mathbf{x} + b_d)$ for some point-wise activation $\sigma(\cdot)$ and parameterized by $\theta_d = (\mathbf{w}_d, b_d) \in \mathbb{R}^s \times \mathbb{R}$, the representation layer reduces to the standard convolutional layer. Other possibilities, e.g. gaussian functions with diagonal covariances, have also been considered in Cohen et al. (2016a). Following the representation layer, are hidden layers indexed by $l = 0, \ldots, L - 1$, each begins with a $1 \times 1$ *conv* operator, which is just an $r_{l-1} \times 1 \times 1$ convolutional layer with $r_{l-1}$ input channels and $r_l$ output channels, with the sole exception that parameters of each kernel could be spatially unshared (known as locally-connected layer (Taigman et al., 2014)). Following each *conv* layer is a spatial pooling, that takes products of non-overlapping two-dimensional windows covering the output of the previous layer, where for $l = L - 1$ the pooling window is the size of the entire spatial dimension (i.e. global pooling), reducing its output's shape to a $r_{L-1} \times 1 \times 1$, i.e. an $r_{L-1}$-dimensional vector. The final $L$ layer maps this vector with a dense linear layer into the $Y$ network outputs, denoted by $\mathbf{h}_y(\mathbf{x}_1, \ldots, \mathbf{x}_N)$, representing score functions classifying each $X$ to one of the classes through: $y^* = \arg\max_y \mathbf{h}_y(\mathbf{x}_1, \ldots, \mathbf{x}_N)$. As shown in Cohen et al. (2016a), these functions have the following form:

$$\mathbf{h}_y(\mathbf{x}_1, \ldots, \mathbf{x}_N) = \sum_{d_1, \ldots, d_N = 1}^{M} \mathcal{A}_{d_1, \ldots, d_N}^y \prod_{i=1}^{N} f_{\theta_{d_i}}(\mathbf{x}_i) \tag{5}$$

where $\mathcal{A}^y$, called the *coefficients tensor*, is a tensor of order $N$ and dimension $M$ in each mode, which for the sake of discussion can simply be seen as a multi-dimensional array, specified by $N$ indices $d_1, \ldots, d_N$ each ranging in $\{1, \ldots, M\}$, with entries given by polynomials in the network's *conv* weights. A byproduct of eq. 5 is that for a fixed set of $M$ representation functions, all functions represented by ConvACs lay in the same subspace of functions.

## B COMPARISON TO POOLING GEOMETRY

From theorem 1 we learn that overlaps give rise to networks which almost always cannot be efficiently implemented by non-overlapping ConvAC with standard pooling geometry. However, as proven by Cohen and Shashua (2017), a ConvAC that uses a different pooling geometry – i.e. the input to the pooling layers are not strictly contiguous windows from the previous layer – also cannot be efficiently implemented by the standard ConvAC with standard pooling geometry. This raises the question of whether overlapping operations are simply equivalent to a ConvAC with a different pooling geometry and nothing more. We answer this question in two parts. First, a ConvAC with a different pooling geometry might be able to implement some function more efficiently than ConvAC with standard pooling geometry, however, the reverse is also true, that a ConvAC with standard pooling can implement some functions more efficiently than ConvAC with alternative pooling. In contrast, a ConvAC that uses overlaps is still capable to implement efficiently any function that a non-overlapping ConvAC with standard pooling can. Second, we can also show that some overlapping architectures are exponentially efficient than any non-overlapping ConvAC regardless of its pooling geometry. This is accomplished by first extending lemma 1 to this case:

**Lemma 2.** *Under the same conditions as lemma 1, if for all partitions $P \uplus Q$ such that $|P| = |Q| = N/2$ it holds that $\text{rank} \left( \llbracket \mathcal{A}(\mathbf{h}_y) \rrbracket_{P,Q} \right) \geq T$, then any non-overlapping ConvAC regardless of its pooling geometry must have at least $T$ channels in its next to last layer to induce the same grid tensor.*

Next, in theorem 2 below show that some overlapping architectures can induce grid tensors whose matricized rank is exponential for any equal partition of its indices, proving they are indeed exponentially more efficient:

**Theorem 2.** *Under the same settings as theorem 1, consider a GC network whose representation layer is followed by a GC layer with local receptive field $H \times H$, stride $1 \times 1$, and $D \geq M$ output channels, whose parameters are "unshared", i.e. unique to each spatial location in the output of the layer as opposed to shared across them, followed by $(L - 1)$ arbitrary GC layers, whose final output is a scalar. For any choice of the parameters, except a null set (with respect to the Lebesgue measure) and for any template vectors such that $F$ is non-singular, then the matricized rank of the induced grid tensor is equal to $M^{\frac{H^2}{2}}$, for any equal partition of the indices. The exact same result holds if the parameters of the first GC layers are "shared" and $D \geq M \cdot H^2$.*

*Proof sketch.* We follow the same steps of our proof of theorem 1, however, we do not construct just one specific overlapping network that attains a rank of $D \geq M \cdot H^2$, for all possible matricizations of the induced grid tensor. Instead, we construct a separate network for each possible matricization. This proves that with respect to the Lebesgue measure over the network's parameters space, separately for each pooling geometry, the set of parameters for which the lower bound does not hold is of measure zero. Since a finite union of zero measured sets is also of measure zero, then the lower bound with respect to all possible pooling geometries holds almost everywhere, which concludes the proof sketch. See app. C.5 for a complete proof. □

It is important to note that though the above theorem shows that pooling geometry on its own is less expressive than overlapping networks with standard pooling, it does not mean that pooling geometry is irrelevant. Specifically, we do not yet know the effect of combining both overlaps and alternative pooling geometries together. Additionally, many times sufficient expressivity is not the main obstacle for solving a specific task, and the inductive bias induced by a carefully chosen pooling geometry could help reduce overfitting.

# C  DEFERRED PROOFS

In this section we present our proofs for the theorems and claims stated in the body of the article.

## C.1  PRELIMINARIES

In this section we lay out the preliminaries required to understand the proofs in the following sections. We begin with a limited introduction to tensor analysis, followed by quoting a few relevant known results relating tensors to ConvACs.

We begin with basic definitions and operations relating to tensors. Let $\mathcal{A} \in \mathbb{R}^{M_1 \otimes \cdots \otimes M_N}$ be a tensor of order $N$ and dimension $M_i$ in each mode $i \in [N]$ (where $[N] = \{1, \ldots, N\}$), i.e. $\mathcal{A}_{d_1,\ldots,d_N} \in \mathbb{R}$ for all $i \in [N]$ and $d_i \in [M_i]$. For tensors $\mathcal{A}^{(1)}$ and $\mathcal{A}^{(2)}$ of orders $N^{(1)}$ and $N^{(2)}$, and dimensions $M_{i_1}^{(1)}$ and $M_{i_2}^{(2)}$ in each of the modes $i_1 \in [N^{(1)}]$ and $i_2 \in [N^{(2)}]$, respectively, we define their tensor product $\mathcal{A}^{(1)} \otimes \mathcal{A}^{(2)}$ as the order $N^{(1)} + N^{(2)}$ tensor, where

$$\left( \mathcal{A}^{(1)} \otimes \mathcal{A}^{(2)} \right)_{d_1,\ldots,d_{N^{(1)}+N^{(2)}}} = \mathcal{A}^{(1)}_{d_1,\ldots,d_{N^{(1)}}} \cdot \mathcal{A}^{(2)}_{d_{N^{(1)}+1},\ldots,d_{N^{(1)}+N^{(2)}}}$$

For a set of vectors $\mathbf{v}^{(1)} \in \mathbb{R}^{M_1}, \ldots, \mathbf{v}^{(N)} \in \mathbb{R}^{M_N}$, the $N$ ordered tensor $\mathcal{A} = \mathbf{v}^{(1)} \otimes \cdots \otimes \mathbf{v}^{(N)}$ is called an *elementary tensor*, or *rank-1 tensor*. More generally, any tensor can be represented as a linear combination of rank-1 tensors, i.e. $\mathcal{A} = \sum_{z=1}^{Z} \mathbf{v}^{(Z,1)} \otimes \cdots \otimes \mathbf{v}^{(Z,1)}$, known as *rank-1 decomposition*, or CP decomposition, where the minimal $Z$ for which this equality holds is knows as the *tensor rank* of $\mathcal{A}$. Given a set of matrices $F^{(1)} \in \mathbb{R}^{M_1' \times M_1}, \ldots, F^{(N)} \in \mathbb{R}^{M_N' \times M_N}$, we denote by $\mathbf{F} = (F^{(1)} \otimes \cdots \otimes F^{(N)})$ the linear transformation from $\mathbb{R}^{M_1 \otimes \cdots \otimes M_N}$ to $\mathbb{R}^{M_1' \otimes \cdots \otimes M_N'}$, such that for any elementary tensor $\mathcal{A}$, with notations as above, it holds that:

$$\mathbf{F}(\mathcal{A}) = F^{(1)}(\mathbf{v}^{(1)}) \otimes \cdots \otimes F^{(N)}(\mathbf{v}^{(N)})$$

$\mathbf{F}(\mathcal{A})$ is defined for a general tensor $\mathcal{A}$ through its rank-1 decomposition comprising elementary tensors and applying $\mathbf{F}$ on each of them, which can be shown to be equivalent to

$$\mathbf{F}(\mathcal{A})_{k_1,\ldots,k_N} = \sum_{d_1=1}^{M_1} \cdots \sum_{d_N=1}^{M_N} \mathcal{A}_{d_1,\ldots,d_N} \prod_{i=1}^{N} F_{k_i,d_i}^{(i)} \tag{6}$$

A central concept in tensor analysis is that of *tensor matricization*. Let $P \cup Q = [N]$ be a disjoint partition of its indices, such that $P = \{p_1, \ldots, p_{|P|}\}$ with $p_1 < \ldots < p_{|P|}$, and $Q = \{q_1, \ldots, q_{|Q|}\}$ with $q_1 < \ldots < q_{|Q|}$. The matricization of $\mathcal{A}$ with respect to the partition $P \cup Q$, denoted by $[\![\mathcal{A}]\!]_{P,Q}$, is the $\left(\prod_{t=1}^{|P|} M_{p_t}\right)$-by-$\left(\prod_{t=1}^{|Q|} M_{q_t}\right)$ matrix holding the entries of $\mathcal{A}$, such that for all $i \in [N]$ and $d_i \in [M_i]$ the entry $A_{d_1,\ldots,d_N}$ is placed in row index $1 + \sum_{t=1}^{|P|}(d_{p_t} - 1)\prod_{t'=t+1}^{|P|} M_{p_{t'}}$ and column index $1 + \sum_{t=1}^{|Q|}(d_{q_t} - 1)\prod_{t'=t+1}^{|Q|} M_{q_{t'}}$. Applying the matricization operator $[\![\cdot]\!]_{P,Q}$ on the tensor product operator results in the *Kronecker Product*, i.e. for an $N$-ordered tensor $\mathcal{A}$, a $K$-ordered tensor $\mathcal{B}$, and the partition $P \cup Q = [N + K]$, it holds that

$$[\![\mathcal{A} \otimes \mathcal{B}]\!]_{P,Q} = [\![\mathcal{A}]\!]_{P\cap[N],Q\cap[N]} \odot [\![\mathcal{B}]\!]_{(P-N)\cap[K],(Q-N)\cap[K]}$$

where $P - N$ and $Q - N$ are simply the sets obtained by subtracting the number $N$ from every element of $P$ or $Q$, respectively. In concrete terms, the Kronecker product for the matrices $A \in \mathbb{R}^{M_1 \times M_2}$ and $B \in \mathbb{R}^{N_1 \times N_2}$ results in the matrix $A \odot B \in \mathbb{R}^{M_1 N_1 \times M_2 N_2}$ holding $A_{ij}B_{kl}$ in row index $(i-1)N_1 + k$ and column index $(j-1)N_2 + l$. An important property of the Kronecker product is that $\mathrm{rank}(A \odot B) = \mathrm{rank}(A) \cdot \mathrm{rank}(B)$. Typically, when wish to compute $\mathrm{rank}([\![\mathcal{A}]\!]_{P,Q})$, we will first decompose it to a Kronecker product of matrices.

For a linear transform $\mathbf{F}$, as defined above in eq.6, and a partition $P \cup Q$, if $F^{(1)}, \ldots, F^{(N)}$ are non-singular matrices, then $\mathbf{F}$ is invertible and the matrix rank of $[\![\mathcal{A}]\!]_{P,Q}$ equals to the matrix rank of $[\![\mathbf{F}(\mathcal{A})]\!]_{P,Q}$ (see proof in Hackbusch (2012)). Finally, we define the concept of *grid tensors*: for a function $f : \mathbb{R}^s \times \cdots \times \mathbb{R}^s \to \mathbb{R}$ and a set of *template vectors* $\mathbf{x}^{(1)}, \ldots, \mathbf{x}^{(M)} \in \mathbb{R}^s$, the $N$-order grid tensor $\mathcal{A}(f)$ is defined by $(\mathcal{A}(f))_{d_1,\ldots,d_N} = f(\mathbf{x}^{(d_1)}, \ldots, \mathbf{x}^{(d_N)})$.

In the context of ConvACs, circuits and the functions they can realize are typically examined through the matricization of the grid tensors they induce. The following is a succinct summary of the relevant known results used in our proofs – for a more detailed discussion, see previous works (Cohen et al., 2016a; Cohen and Shashua, 2016; 2017). Using the same notations from eq. 5 describing a general ConvAC, let $\mathcal{A}^y$ be the coefficients tensor of order $N$ and dimension $M$ in each mode, and let $f_{\theta_1}, \ldots, f_{\theta_M}:\mathbb{R}^s \to \mathbb{R}$ be a set of $M$ representation functions (see app. A). Under the above definitions, a non-overlapping ConvAC can be said to decompose the coefficients tensor $\mathcal{A}^y$. Different network architectures correspond to known tensor decompositions: shallow networks corresponds to rank-1 decompositions, and deep networks corresponds to Hierarchical Tucker decompositions.

In Cohen and Shashua (2017), it was found that the matrix rank of the matricization of the coefficients tensors $\mathcal{A}^y$ could serve as a bound for the size of networks decomposing $\mathcal{A}^y$. For the conventional non-overlapping ConvAC and the contiguous "low-high" partition $P = \{1, \ldots, N/2\}, Q = \{N/2 + 1, \ldots, N\}$ of $[N]$, the rank of the matricization $[\![\mathcal{A}^y]\!]_{P,Q}$ serves as a lower-bound on the number of channels of the next to last layer of any network which decomposes the coefficients tensor $\mathcal{A}$. In the common case of square inputs, i.e. the input is of shape $H \times H$ and $N = H^2$, it is more natural to represent indices by pairs $(j, i)$ denoting the spatial location of each "patch" $\mathbf{x}_{(j,i)}$, where the first argument denotes the vertical location and the second denotes the horizontal location. Under such setting the equivalent "low-high" partitions are either the "left-right" partition, i.e. $P = \{(j,i)|j \le \frac{H}{2}\}, Q = \{(j,i)|j > \frac{H}{2}\}$, or the "top-bottom" partition, i.e. $P = \{(j,i)|i \le \frac{H}{2}\}, Q = \{(j,i)|i > \frac{H}{2}\}$. More generally, when considering networks using other *pooling geometries*, i.e. not strictly contiguous pooling windows, then for each pooling geometry there exists a corresponding partition $P \cup Q$ such that $\mathrm{rank}([\![\mathcal{A}^y]\!]_{P,Q})$ serves as its respective lower-bound.

Though the results in Cohen and Shashua (2017) are strictly based on the matricization rank of the coefficients tensors, they can be transferred to the matricization rank of grid tensors as well. Grid tensors were first considered for analyzing ConvACs in Cohen and Shashua (2016). For a set of $M$ template vectors $\mathbf{x}^{(1)}, \ldots, \mathbf{x}^{(M)} \in \mathbb{R}^s$, we define the matrix $F \in \mathbb{R}^{M \times M}$ by $F_{ij} = f_{\theta_j}(\mathbf{x}^{(i)})$. With the above notations in place, we can write the the grid tensor $\mathcal{A}(\mathbf{h}_y)$ for the function $\mathbf{h}_y(\mathbf{x}_1, \ldots, \mathbf{x}_N)$ as:

$$\mathcal{A}(\mathbf{h}_y)_{k_1,\ldots,k_N} = \mathbf{h}_y(\mathbf{x}^{(k_1)}, \ldots, \mathbf{x}^{(k_N)})$$
$$= \sum_{d_1,\ldots,d_N=1}^{M} \mathcal{A}^y_{d_1,\ldots,d_N} \prod_{i=1}^{N} f_{\theta_{d_i}}(\mathbf{x}^{(k_i)})$$
$$= \sum_{d_1,\ldots,d_N=1}^{M} \mathcal{A}^y_{d_1,\ldots,d_N} \prod_{i=1}^{N} F_{k_i,d_i}$$
$$\Rightarrow \mathcal{A}(\mathbf{h}_y) = (F \otimes \cdots \otimes F)(\mathcal{A}^y)$$

If the representation functions are linearly independent and continuous, then we can choose the template vectors such that $F$ is non-singular (see Cohen and Shashua (2016)), which according to the previous discussion on tensor matricization, means that for any partition $P \cup Q$ and any coefficients tensor $\mathcal{A}^y$, it holds that $\mathrm{rank}([\![\mathcal{A}^y]\!]_{P,Q}) = \mathrm{rank}([\![\mathcal{A}(\mathbf{h}_y)]\!]_{P,Q})$. Thus, any lower bound on the matricization rank of the grid tensor

translates to a lower bound on the matricization rank of the coefficients tensors, which in turn serves as lower bound on the size of non-overlapping ConvACs. The above discussion leads to the proof of lemma 1 and lemma 2 that were previously stated:

*Proof of lemma 1 and lemma 2.* For the proofs of the base results with respect to the coefficients tensor, see Cohen and Shashua (2017). To prove it is possible to choose the template vectors such that $F$ is non-singular, see Cohen and Shashua (2016). To prove that if $F$ is non-singular, then the grid tensor and the coefficients tensor have the same matricization rank, see lemma 5.6 in Hackbusch (2012). $\square$

We additionally quote the following lemma regarding the prevalence of the maximal matrix rank for matrices whose entries are polynomial functions:

**Lemma 3.** *Let $M, N, K \in \mathbb{N}$, $1 \leq r \leq \min\{M, N\}$ and a polynomial mapping $A : \mathbb{R}^K \to \mathbb{R}^{M \times N}$, i.e. for every $i \in [M]$ and $j \in [N]$ it holds that $A_{ij} : \mathbb{R}^k \to \mathbb{R}$ is a polynomial function. If there exists a point $\mathbf{x} \in \mathbb{R}^K$ such that $\mathrm{rank}\,(A(\mathbf{x})) \geq r$, then the set $\{\mathbf{x} \in \mathbb{R}^K | rank A(\mathbf{x}) < r\}$ has zero measure (with respect to the Lebesgue measure over $\mathbb{R}^K$).*

*Proof.* See Sharir et al. (2016). $\square$

Finally, we simplify the notations of the GC layer with product pooling function for the benefit of following our proofs below. We will represent the parameters of the $l$'th GC layer by $\{(\mathbf{w}^{(l,c)} \in \mathbb{R}^{D^{(l-1)} \times R^{(l)} \times R^{(l)}}, \mathbf{b}^{(l,c)} \in \mathbb{R}^{R^{(l)} \times R^{(l)}})\}_{c \in [D^{(l)}]}$, where $\mathbf{w}^{(l,c)}$ represents the weights and $\mathbf{b}^{(l,c)}$ the biases. Let $X \in \mathbb{R}^{H^{(in)} \times H^{(in)} \times D^{(in)}}$ be the input to the layer and $Y \in \mathbb{R}^{D^{(out)} \times H^{(out)} \times H^{(out)}}$ be the output, then the following equality holds:

$$Y_{c,u,v} = \prod_{j=1}^{R^{(l)}} \prod_{i=1}^{R^{(l)}} \left( b_{ji}^{(l,c)} + \sum_{d=1}^{D^{(in)}} w_{dji}^{(l,c)} X_{d,uS^{(l)}+j,vS^{(l)}+i} \right) \tag{7}$$

The above treats the common case where the parameters are shared across all spatial locations, but sometimes we wish to consider the "unshared" case, in which there is are different weights and biases for each location, which we denote by $\{(\mathbf{w}^{(l,c,u,v)} \in \mathbb{R}^{D^{(l-1)} \times R^{(l)} \times R^{(l)}}, \mathbf{b}^{(l,c,u,v)} \in \mathbb{R}^{R^{(l)} \times R^{(l)}})\}_{c \in [D^{(l)}], u \in [H^{(out)}], v \in [H^{(out)}]}$.

With the above definitions and lemmas in place, we are ready to prove the propositions and theorems from the body of the article.

## C.2 Proof of Proposition 1

Proposition 1 is a direct corollary of the following two claims:

**Claim 1.** *Let $f : \mathbb{R}^{D^{(in)} \times H^{(in)} \times H^{(in)}} \to \mathbb{R}^{D^{(out)} \times H^{(out)} \times H^{(out)}}$ be a function realized by a single GC layer with $R \times R$ local receptive field, $S \times S$ stride, and $D^{(out)}$ output channels, that is parameterized by $\{(\mathbf{w}^{(c)}, \mathbf{b}^{(c)})\}_{c=1}^{D^{(out)}}$. For all $\tilde{R} \geq R$, a GC layer with $\tilde{R} \times \tilde{R}$ local receptive field, $S \times S$ stride, and $C$ output channels, parameterized by $\{(\tilde{\mathbf{w}}^{(c)}, \tilde{\mathbf{b}}^{(c)})\}_{c=1}^{D^{(out)}}$, could also realize $f$. The same is true for the unshared case of both layers.*

*Proof.* The claim is trivially satisfied by setting $\tilde{\mathbf{w}}^{(c)}$ such that it is equal to $\mathbf{w}^{(c)}$ in all matching coordinates, while using zeros for all other coordinates. Similarly, we set $\tilde{\mathbf{b}}^{(c)}$ to be equal to $\mathbf{b}^{(c)}$ in all matching coordinates, while using ones for all other coordinates. $\square$

**Claim 2.** *Let $f : \mathbb{R}^{D^{(in)} \times H^{(in)} \times H^{(in)}} \to \mathbb{R}^{D^{(out)} \times H^{(out)} \times H^{(out)}}$ be a function realized by a GC layer with $R \times R$ local receptive field and $1 \times 1$ stride, parameterized by $\{(\mathbf{w}^{(c)}, \mathbf{b}^{(c)})\}_{c=1}^{D^{(out)}}$. Then there exists an assignment to $(\mathbf{w}, \mathbf{b})$ such that $f$ is the identity function $f(X) = X$. The same is true for the unshared case of both layers.*

*Proof.* From claim 1 it is sufficient to show the above holds for $R = 1$. Indeed, setting

$$w_d^{(c)} = 1_{[d=c]} = \begin{cases} 1 & d = c \\ 0 & d \neq c \end{cases}$$

and $\mathbf{b}^{(c)} \equiv \mathbf{0}$ satisfies the claim. $\square$

### C.3 PROOF OF THEOREM 1

We wish to show that for all choices of parameters, except a null set (with respect to Lebesgue measure) the grid tensor induced by the given GC network has rank satisfying eq. 4. Since the entries of the matricized grid tensor are polynomial function of its parameters, then according to lemma 3, it is sufficient to find a single example that achieves this bound. Hence, our proof is simply the construction of such an example.

Recall that the template vectors must hold that for the matrix $F$, defined by $F_{ij} = f_j(\mathbf{x}^{(i)})$, where $\{f_j\}_{j=1}^M$ are the representation matrices, is a non-singular matrix. We additionally assume in the following claims that the output of the representation layer is of width and height equal to $H \in \mathbb{N}$, where $H$ is an even number – the claims and proofs can however be easily adapted to the more general case.

Assume a ConvAC as described in the theorem, with representation layer defined according to above, followed by $L$ GC layers, where the $l$'th layer has a local receptive field of $R^{(l)}$, a stride of $S^{(l)}$, and $D^{(l)}$ output channels. We first construct our example, that achieves the desired matricization rank, for the simpler case where the first layer following the representation layer has a local receptive field large enough, i.e. when it is larger than $\frac{H}{2}$. Recall that for the first layer the total receptive field is equal to its local receptive field. In the context of theorem 1, this first layer satisfies the conditions necessary to produce the lower bound given in the theorem.

The specific construction is presented in the following claim, which relies on utilizing the large local receptive field to match each spatial location in the left side of the input with one from the right side, such that for each such pair, the respective output of the first layer will represent a mostly diagonal matrix. We then set the rest of the parameters such that the output of the entire network is defined by a tensor product of mostly diagonal matrices. Since the matricization rank of the tensor product of matrices is equal to the product of the individual ranks, it results in an exponential form of the rank as is given in the theorem.

**Claim 3.** *Assume a ConvAC as defined above, ending with a single scalar output. For all $l \in [L]$, the parameters of the $l$-th GC layer are denoted by $\{(\mathbf{w}^{(l,c)} \in \mathbb{R}^{D^{(l-1)} \times R^{(l)} \times R^{(l)}}, \mathbf{b}^{(l,c)} \in \mathbb{R}^{R^{(l)} \times R^{(l)}})\}_{c \in [D^{(l)}]}$. Let $\mathbf{h}(\mathbf{x}_1, \ldots, \mathbf{x}_N)$ be the function realized the output of network. Additionally define $R \equiv R^{(1)}$, $S \equiv S^{(1)}$ and $D \equiv \min\{D^{(1)}, M\}$. If $R > \frac{H}{2}$, and the weights $\mathbf{w}^{(1,c)}$ and biases $\mathbf{b}^{(1,c)}$ of the first GC layer layer are set to:*

$$w_{mji}^{(1,c)} = \begin{cases} -\alpha \left(F^{-1}\right)_{m,c} & c \leq D \text{ and } (j,i) \in \{(1,1), (\rho,\tau)\} \\ 0 & \textit{Otherwise} \end{cases}$$

$$b_{ji}^{(1,c)} = \begin{cases} \beta & c \leq D \text{ and } (j,i) \in \{(1,1), (\rho,\tau)\} \\ 1 & c \leq D \text{ and } (j,i) \notin \{(1,1), (\rho,\tau)\} \\ 0 & \textit{Otherwise} \end{cases}$$

*where $\beta = \frac{2\alpha}{D}$, then there exists an assignment to $\alpha$ and the parameters of the other GC layers such that $\mathrm{rank}\left([\![\mathcal{A}(\mathbf{h})]\!]_{P,Q}\right) = D^{\lfloor \frac{H-R}{S}+1 \rfloor \cdot \lceil \frac{H}{S} \rceil}$, where $P \cup Q$ is either the "left-right" or "top-bottom" partition, and $(\rho, \tau)$ equals to $(1, R)$ or $(R, 1)$, respectively.*

*Proof.* The proof for either the "left-right" or "top-bottom" partition is completely symmetric, thus it is enough to prove the claim for the "left-right" case, where $(\rho, \tau) = (1, R)$. We wish to compute the entry $\mathcal{A}(\mathbf{h})_{d_{(1,1)}, \ldots, d_{(H,H)}}$ of the induced grid tensor for arbitrary indices $d_{(1,1)}, \ldots, d_{(H,H)}$. Let $O \in \mathbb{R}^{M \times H \times H}$ be the 3-order tensor output of the representation layer, where $O_{m,j,i} = F_{d_{(j,i)},m}$ for the aforementioned indices and for all $1 \leq i, j \leq H$ and $m \in [M]$.

We begin by setting the parameters of all layers following the first GC layer, such that they are equal to computing the sum along the channels axis of the output of the first GC layer, followed by a global product of all of the resulting sums. To achieve this, we can first assume w.l.o.g. that these layers are non-overlapping through proposition 1. We then set the parameters of the second GC layer to $\mathbf{w}^{(2,c)} = \mathbf{1}$ and $\mathbf{b}^{(2,c)} \equiv \mathbf{0}$, i.e. all ones and all zeros, respectively, which is equivalent to taking the sum along the channels axis for each spatial location, followed by taking the products over non-overlapping local receptive fields of size $R^{(2)}$. For the other layers, we simply set them as to take just the output of the first channel of the output of the preceding layer, which is equal to setting their parameters to $w_{dji}^{(l,c)} = \begin{cases} 1 & d = 1 \\ 0 & d \neq 1 \end{cases}$ and $\mathbf{b}^{(l,c)} \equiv \mathbf{0}$. Setting the parameters as described above results in:

$$(\mathcal{A}(\mathbf{h}))_{d_{(1,1)}, \ldots, d_{(H,H)}} = \prod_{\substack{0 \leq uS < H \\ 0 \leq vS < H}} \sum_{c=1}^{D^{(1)}} \prod_{j,i=1}^{R} \underbrace{\left( b_{ji}^{(1,c)} + \sum_{m=1}^{M} w_{mji}^{(1,c)} O_{m,uS+j,vS+i} \right)}_{g(u,v,c,j,i)} \tag{8}$$

where we extended $O$ with zero-padding for the cases where $uS + j > H$ or $vS + i > H$, as mentioned in sec. 2. Next, we go through the technical process of reducing eq. 8 to a product of matrices.

Substituting the values of $w_{dji}^{(1,c)}$ and $b_{ji}^{(1,c)}$ with those defined in the claim, and computing the value of $g(u, v, c, j, i)$ results in:

$$g(u, v, c, j, i) = \begin{cases} \beta - \alpha \sum_{m=1}^{M} \left(F^{-1}\right)_{m,c} F_{d_{(uS+j,vS+i)},m} & c \le D \text{ and } vS + R \le H \text{ and } (j,i) \in \{(1,1),(1,R)\} \\ \beta - \alpha \sum_{m=1}^{M} \left(F^{-1}\right)_{m,c} F_{d_{(uS+j,vS+i)},m} & c \le D \text{ and } vS + R > H \text{ and } (j,i) = (1,1) \\ \beta & c \le D \text{ and } vS + R > H \text{ and } (j,i) = (1,R) \\ 1 & c \le D \text{ and } (j,i) \notin \{(1,1),(1,R)\} \\ 0 & c > D \end{cases}$$

$$= \begin{cases} \beta - \alpha \left(FF^{-1}\right)_{d_{(uS+1,vS+i)},c} & c \le D \text{ and } vS + R \le D \text{ and } (j,i) \in \{(1,1),(1,R)\} \\ \beta - \alpha \left(FF^{-1}\right)_{d_{(uS+1,vS+i)},c} & c \le D \text{ and } vS + R > H \text{ and } (j,i) = (1,1) \\ \beta & c \le D \text{ and } vS + R > H \text{ and } (j,i) = (1,R) \\ 1 & c \le D \text{ and } (j,i) \notin \{(1,1),(1,R)\} \\ 0 & c > D \end{cases}$$

from which we derive:

$$f(u,v) \equiv \sum_{c=1}^{D^{(1)}} \prod_{j,i=1}^{R} g(u,v,c,j,i) = \begin{cases} D\beta^2 - \alpha\beta(1_{[d_{(uS+1,vS+1)} \le D]} + 1_{[d_{(uS+1,vS+R)} \le D]}) & vS + R \le H \\ \quad + \alpha^2 1_{[d_{(uS+1,vS+1)} = d_{(uS+1,vS+R)} \le D]} \\ D\beta^2 - \alpha\beta 1_{[d_{(uS+1,vS+1)} \le D]} & vS + R > H \end{cases}$$

where $(\mathcal{A}(\mathbf{h}))_{d_{(1,1)},\dots,d_{(H,H)}} = \prod_{\substack{0 \le uS < H \\ 0 \le vS < H}} f(u,v)$.

At this point we branch into two cases. If $S$ divides $R - 1$, then for all $u, v \in \mathbb{N}$ such that $vS + R \le H$ and $uS < H$, the above expression for $f(u,v)$ and $f(u, v + \frac{R-1}{S})$ depends only on the indices $d_{(uS+1,vS+1)}$ and $d_{(uS+1,vS+R)}$, while these two indices affect only the aforementioned expressions. By denoting $A_{d_{(uS+1,vS+1)},d_{(uS+1,vS+R)}}^{(u,v)} = f(u,v) \cdot f(u, v + \frac{R-1}{S})$, we can write it as:

$$A_{ij}^{(u,v)} = \begin{cases} \left(D\beta^2 - 2\alpha\beta + \alpha^2 1_{[i=j]}\right)\left(D\beta^2 - \alpha\beta\right) & i, j \le D \\ \left(D\beta^2 - \alpha\beta\right)\left(D\beta^2\right) & i \le D \text{ and } j > D \\ \left(D\beta^2 - \alpha\beta\right)^2 & i > D \text{ and } j \le D \\ \left(D\beta^2\right)^2 & i, j > D \end{cases}$$

where $i, j \in [M]$ stand for the possible values of $d_{(uS+1,vS+1)}$ and $d_{(uS+1,vS+R)}$, respectively. Substituting $\beta = \frac{2\alpha}{D}$, as stated in the claim, and setting $\alpha = \left(\frac{D}{2}\right)^{1/4}$, results in:

$$A_{ij}^{(u,v)} = \begin{cases} 1_{[i=j]} & i, j \le D \\ \frac{4}{D} & i \le D \text{ and } j > D \\ \frac{2}{D} & i > D \text{ and } j \le D \\ \frac{8}{D} & i, j > D \end{cases}$$

which means $\text{rank}\left(A^{(u,v)}\right) = D$. Since $(\mathcal{A}(\mathbf{h}))_{d_{(1,1)},\dots,d_{(H,H)}}$ equals to the product $\prod_{\substack{0 \le uS < H \\ 0 \le vS \le H-R}} A_{d_{(uS+1,vS+1)},d_{(uS+1,vS+R)}}^{(u,v)}$, then $[\![\mathcal{A}(\mathbf{h})]\!]_{P,Q}$ equals to the Kronecker product of the matrices in $\{A^{(u,v)} | 0 \le uS < H, 0 \le vS \le H - R\}$, up to permutation of its rows and columns, which do not affect its matrix rank. Thus, the matricization rank of $\mathcal{A}(\mathbf{h})$ satisfies:

$$\text{rank}\left([\![\mathcal{A}(\mathbf{h})]\!]_{P,Q}\right) = D^{\left|\{A^{(u,v)} | 0 \le uS < H, 0 \le vS \le H-R\}\right|} = D^{\left\lfloor \frac{H-R}{S}+1\right\rfloor \cdot \left\lceil \frac{H}{S}\right\rceil}$$

which proves the claim for this case.

If $S$ does not divide $R - 1$, then for all $u, v \in \mathbb{N}$, such that $vS + R \le H$ and $uS < H$, it holds that $f(u,v)$ depends only on the indices $d_{(uS+1,vS+1)}$ and $d_{(uS+1,vS+R)}$, and they affect only $f(u,v)$. Additionally, for all $u, v \in \mathbb{N}$, such that $H < vS + R$, $vS < H$ and $uS < H$, it holds that $f(u,v)$ depends only on the index $d_{(uS+1,vS+1)}$, and this index affects only $f(u,v)$. Let us denote $A_{d_{(uS+1,vS+1)},d_{(uS+1,vS+R)}}^{(u,v)} = f(u,v)$ for $vS + R \le H$:

$$A_{ij}^{(u,v)} = \begin{cases} \left(D\beta^2 - 2\alpha\beta + \alpha^2 1_{[i=j]}\right) & i, j \le D \\ \left(D\beta^2 - \alpha\beta\right) & i \le D \text{ and } j > D \\ \left(D\beta^2 - \alpha\beta\right) & i > D \text{ and } j \le D \\ \left(D\beta^2\right) & i, j > D \end{cases}$$

which by setting $\beta = \frac{2\alpha}{D}$ and $\alpha = 1$ results in:

$$A_{ij}^{(u,v)} = \begin{cases} 1_{[i=j]} & i,j \leq D \\ \frac{2}{D} & i \leq D \text{ and } j > D \\ \frac{2}{D} & i > D \text{ and } j \leq D \\ \frac{4}{D} & i,j > D \end{cases}$$

which means $\text{rank}\left(A^{(u,v)}\right) = D$. For $vS + R > H$, and we can define the vector $\mathbf{a}_{d_{(uS+1,vS+1)}}^{(u,v)} = f(u,v)$, which by using the same values of $\alpha$ and $\beta$ results in:

$$a_i^{(u,v)} = \begin{cases} \frac{2}{D} & i \leq D \\ \frac{4}{D} & i > D \end{cases}$$

By viewing $\mathbf{a}^{(u,v)}$ as either a column or row vector, depending on whether $d_{(uS+1,vS+1)} \in P$ or $d_{(uS+1,vS+1)} \in Q$, respectively, it holds that $[\![\mathcal{A}(\mathbf{h})]\!]_{P,Q}$ equals to the Kronecker product of the matrices in $\{A^{(u,v)}\}_{\substack{0 \leq uS < H \\ 0 \leq vS \leq H-R}} \cup \{\mathbf{a}^{(u,v)}\}_{\substack{0 \leq uS < H \\ H-R < vS < H}}$, up to permutations of its rows and columns, which do not affect the rank. Since $\mathbf{a}^{(u,v)} \neq \mathbf{0}$ then $\text{rank}\left(\mathbf{a}^{(u,v)}\right) = 1$, which means the matricization rank of $\mathcal{A}(\mathbf{h})$ once again holds:

$$\text{rank}\left([\![\mathcal{A}(\mathbf{h})]\!]_{P,Q}\right) = D^{\left|\{A^{(u,v)}|0 \leq uS < H, 0 \leq vS \leq H-R\}\right|} = D^{\left\lfloor \frac{H-R}{S}+1 \right\rfloor \cdot \left\lceil \frac{H}{S} \right\rceil}$$

$\square$

In the preceding claim we have describe an example for the case where the total receptive of the first GC layer is already large enough for satisfying the conditions of the theorem. In the following claim we extend this result for the general case. This is accomplished by showing that a network comprised of just $L$ GC layers with local receptive fields $\{R^{(l)} \times R^{(l)}\}_{l \in [L]}$, strides $\{S^{(l)} \times S^{(l)}\}_{l \in [L]}$, and output channels $\{D^{(l)}\}_{l \in [L]}$, can effectively compute the same output as the first GC layer from claim 3, for all inputs [3].

Recall that the layer from claim 3 performs an identical transformation on each $M \times 1 \times 1$ patch from its input, followed by taking the point-wise product of far-away pairs of transformed patches. Thus, the motivation behind the specific construction we use, is to use the first of the $L$ layers to perform this transformation, while using half of its output channels for storing the transformed patch from the same location, and the other half for storing a transformed patch, but from a location farthest to the right, constrained by its local receptive field. This is equal to having one set of transformed patches sitting still, while another "shifted" set of transformed patches. The other layers simply pass the the first half of the channels as is, using an identity operation as defined in claim 2, while continuously shifting the other half of the channels more and more to the left, bringing faraway patches closer together. Finally, at the last layer we take both halves and multiple them together.

**Claim 4.** *Assume $\Phi$ is a ConvAC comprised of just $L$ GC layers as described above, where the output of the network is not limited to a scalar value. Assume the total stride of the $L$-th GC layer is greater than $H/2$, and let $T_S^{(L)}$ and $T_R^{(L,\alpha)}$ be the total stride and the $\alpha$-minimal total receptive field, respectively, for $\alpha = \lfloor H/2 \rfloor + 1$. Let $\Psi$ be a ConvAC comprised of a single GC layer with local receptive field $R \equiv T_R^{(L,\alpha)}$, stride $S \equiv T_S^{(L)}$, output channels $D \equiv \min\{\frac{1}{2}\min_{1 \leq l < L} D^{(l)}, D^{(L)}\}$, where its weights and biases are set to the following:*

$$w_{mji}^{(c)} = \begin{cases} A_{m,c} & (j,i) \in \{(1,1),(\rho,\tau)\} \\ 0 & \text{Otherwise} \end{cases}$$

$$b_{ji}^{(c)} = \begin{cases} \beta_c & (j,i) \in \{(1,1),(\rho,\tau)\} \\ 1 & \text{Otherwise} \end{cases}$$

*for $\beta \in \mathbb{R}^D$, $A \in \mathbb{R}^{M \times D}$ and $(\rho,\tau) \in \{(1,R),(R,1)\}$. Then, there exists a set of weights to the layers of $\Phi$ such that for every input $X$, the output of $\Phi$ is equivalent to the output of $\Psi$ for channels $\leq D$, and zero otherwise.*

*Proof.* The two possible cases for $(\rho,\tau)$ are completely symmetric, thus it is enough to prove the claim just for $(\rho,\tau) = (1,R)$. Additionally, we can assume w.l.o.g. that $\forall l, R^{(l)} > 1$, by setting any $1 \times 1$ layer to act as pass-through according to claim 2, and also assume that the $\alpha$-minimal total receptive field is exactly equal to the total receptive field of the $L$-th layer, by applying claim 1 to realize an equivalent network with smaller windows. Finally, the case for $L = 1$ is trivial, and thus we assume $L > 1$.

---

[3] Notice that in this context, there is no representation layer, and the input can be any 3-order tensor.

Let us set the parameters $\left\{ \mathbf{w}^{(l,k)}, \mathbf{b}^{(l,k)} \right\}$ of the layers of $\Phi$ as follows:

$$
w_{dji}^{(l,k)} = \begin{cases}
-A_{d,k} & (l=1) \text{ and } (1 \le k \le D) \text{ and } (j,i) = (1,1) \\
-A_{d,k-D} & (l=1) \text{ and } (D < k \le 2D) \text{ and } (j,i) = (1, R^{(l)}) \\
1_{[d=k]} & (1 < l < L) \text{ and } (1 \le k \le D) \text{ and } (j,i) = (1,1) \\
1_{[d=k]} & (1 < l < L) \text{ and } (D < k \le 2D) \text{ and } (j,i) = (1, R^{(l)}) \\
1_{[d=k]} & (l=L) \text{ and } (1 \le k \le D) \text{ and } (j,i) = (1,1) \\
1_{[d=M+k]} & (l=L) \text{ and } (1 \le k \le D) \text{ and } (j,i) = (1, R^{(l)}) \\
0 & \text{Otherwise}
\end{cases}
$$

$$
b_{ji}^{(l,k)} = \begin{cases}
\beta_k & (l=1) \text{ and } (1 \le k \le D) \text{ and } (j,i) = (1,1) \\
\beta_{k-D} & (l=1) \text{ and } (D < k \le 2D) \text{ and } (j,i) = (1, R^{(l)}) \\
0 & (1 < l < L) \text{ and } (1 \le k \le D) \text{ and } (j,i) = (1,1) \\
0 & (1 < l < L) \text{ and } (D < k \le 2D) \text{ and } (j,i) = (1, R^{(l)}) \\
0 & (l=L) \text{ and } (1 \le k \le D) \text{ and } (j,i) = (1,1) \\
0 & (l=L) \text{ and } (1 \le k \le D) \text{ and } (j,i) = (1, R^{(l)}) \\
1 & \text{Otherwise}
\end{cases}
$$

It is left to prove the above satisfies the claim. Let $O^{(l)} \in \mathbb{R}^{D^{(l)} \times H^{(l)} \times H^{(l)}}$ be the output of the $l$-th layer, for $l \in [0, \dots, L]$, where $H^{(l)}$ is the width and height of the output. We additionally assume that for indices beyond the bounds of $[D^{(l)}] \times [H^{(l)}] \times [H^{(l)}]$ the value of $O^{(l)}$ is zero, i.e. we assume zero padding when applying the convolutional operation of the GC layer. We extend the definition for $l = 0$, by setting $D^{(0)} \equiv M$ and $H^{(0)} \equiv H$, where we identify $O^{(0)}$ with the input to the network $\Phi$. Given the above, the output of the first layer for $k \in [D^{(1)}]$ and $0 \le u, v < H^{(1)}$, is as follows:

$$
O_{k,u+1,v+1}^{(1)} = \prod_{j,i=1}^{R^{(1)}} \left( b_{ji}^{(1,k)} + \sum_{d=1}^{D^{(0)}} w_{dji}^{(1,k)} O_{d,uS_h+j,vS_w+i}^{(0)} \right)
$$

$$
= \begin{cases}
\beta_k + \sum_{d=1}^{M} A_{d,k} \cdot O_{d,uS_h+1,vS_w+1}^{(0)} & 1 \le k \le D \\
\beta_{k-D} + \sum_{d=1}^{M} A_{d,k-D} \cdot O_{d,uS_h+1,vS_w+R^{(1)}}^{(0)} & D < k \le 2D \\
1 & \text{Otherwise}
\end{cases}
$$

We will show by induction that for $1 < l < L$, $k \in [D^{(l)}]$ and $0 \le u, v < H^{(l)}$ the output of the $l$-th layer $O_{k,u+1,v+1}^{(l)}$ always equals to:

$$
O_{k,u+1,v+1}^{(l)} = \begin{cases}
O_{k,u\eta^{(l)}+1,v\eta^{(l)}+1}^{1} & k \le D \\
O_{k,u\eta^{(l)}+1,v\eta^{(l)}+\xi^{(l)}}^{1} & D < k \le 2D \\
1 & \text{Otherwise}
\end{cases}
$$

where $\eta^{(l)} = \prod_{i=2}^{l} S^{(i)}$ and $\xi^{(l)} = R^{(l)} \cdot \eta^{(l-1)} + \sum_{k=2}^{l-1} (R^{(k)} - S^{(l)}) \cdot \eta^{(k-1)}$. It is trivial to verify that for $l = 2$ it indeed holds, since:

$$
O_{k,u+1,v+1}^{(2)} = \begin{cases}
O_{k,uS^{(2)}+1,vS^{(2)}+1}^{(1)} & k \le D \\
O_{k,uS^{(2)}+1,vS^{(2)}+R^{(2)}}^{(1)} & D < k \le 2D \\
1 & \text{Otherwise}
\end{cases}
$$

where $\eta^{(2)} = S^{(2)}$ and $\xi^{(2)} = R^{(2)}$. Assume the claim holds up to $l - 1$, and we will show it also holds for $l$:

$$O^{(l)}_{k,u+1,v+1} = \prod_{j,i=1}^{R^{(l)}} \left( b^{(l,k)}_{ji} + \sum_{d=1}^{D^{(l-1)}} w^{(l,k)}_{dji} O^{l-1}_{d,uS^{(l)}+j,vS^{(l)}+i} \right)$$

$$= \begin{cases} O^{(l-1)}_{k,uS^{(l)}+1,vS^{(l)}+1} & k \leq D \\ O^{(l-1)}_{k,uS^{(l)}+1,vS^{(l)}+R^{(l)}} & D < k \leq 2D \\ 1 & \text{Otherwise} \end{cases}$$

$$\text{Induction Hypothesis} \Rightarrow = \begin{cases} O^{(1)}_{k,\left(uS^{(l)}\right)\eta^{(l-1)}+1,\left(vS^{(l)}\right)\eta^{(l-1)}+1} & k \leq D \\ O^{(1)}_{k,\left(uS^{(l)}\right)\eta^{(l-1)}+1,\left(vS^{(l)}+R^{(l)}-1\right)\eta^{(l-1)}+\xi^{(l-1)}} & D < k \leq 2D \\ 1 & \text{Otherwise} \end{cases}$$

$$= \begin{cases} O^1_{k,u\eta^{(l)}+1,v\eta^{(l)}+1} & k \leq D \\ O^1_{k,u\eta^{(l)}+1,v\eta^{(l)}+\xi^{(l)}} & D < k \leq 2D \\ 1 & \text{Otherwise} \end{cases}$$

Were we used the fact that $\eta^{(l)} = S^{(l)}\eta^{(l-1)}$ and $\xi^{(l)} = R^{(l)}\eta^{(l-1)} + \xi^{(l-1)} - \eta^{(l-1)}$.

Finally, we show that $O^{(L)}_{k,u+1,v+1}$ for $k \leq D$ and $0 \leq u, v < H^{(L)}$ equals to the output of the single GC layer specified in the claim:

$$O^{(L)}_{k,u+1,v+1} = \prod_{j,i=1}^{R^{(L)}} \left( b^{(L,k)}_{ji} + \sum_{d=1}^{D^{(L-1)}} w^{(L,k)}_{dji} O^{(L-1)}_{d,uS^{(L)}+j,vS^{(L)}+i} \right)$$

$$= O^{(L-1)}_{k,uS^{(L)}+1,vS^{(L)}+1} \cdot O^{(L-1)}_{k,uS^{(L)}+1,vS^{(L)}+R^{(L)}}$$

$$= O^{(1)}_{k,u\eta^{(L)}+1,v\eta^{(L)}+1} \cdot O^{(1)}_{k,u\eta^{(L)}+1,v\eta^{(L)}+\xi^{(L)}}$$

$$= \left( \beta_k + \sum_{d=1}^{D^{(0)}} A_{d,k} O^{(0)}_{d,u\eta^{(L)}S^{(1)}+1,v\eta^{(L)}S^{(1)}+1} \right)$$

$$\cdot \left( \beta_k + \sum_{d=1}^{D^{(0)}} A_{d,k} O^{(0)}_{d,u\eta^{(L)}S^{(1)}+1,\left(v\eta^{(L)}+\xi^{(L)}-1\right)S^{(1)}+R^{(1)})} \right)$$

$$= \left( \beta_k + \sum_{d=1}^{D^{(0)}} A_{d,k} O^{(0)}_{d,uT^{(L)}_S+1,vT^{(L)}_S+1} \right) \left( \beta_k + \sum_{d=1}^{D^{(0)}} A_{d,k} O^{(0)}_{d,uT^{(L)}_S+1,vT^{(L)}_S+T^{(L)}_R} \right)$$

which is indeed equal to the single GC layer. For $k > D$, both the bias and the weights for the last layer are zero, and thus $O^{(L)}_{k,u+1,v+1} = 1$. $\qquad\square$

Finally, with the above two claims set in place, we can prove our main theorem:

*Proof.* (**of theorem 1**) Using claim 4 we can realize the networks from claim 3, for which the matricization rank for either partition equals to:

$$D^{\left\lfloor \frac{H - T_R^{(K,\lfloor H/2\rfloor)}}{T_S} + 1 \right\rfloor \cdot \left\lceil \frac{H}{T_S} \right\rceil}$$

Since for any matricization $[\mathcal{A}(\Psi)]_{P,Q}$ the entries of the matricization are polynomial functions with respect to the parameters of the network, then, according to lemma 3, the set of parameters of $\Phi$, that does not attain the above rank, has zero measure. Since the union of zero measured sets is also of measure zero, then all parameters except a set of zero measure attain this matricization rank for both partitions at once, concluding the proof. $\qquad\square$

## C.4 PROOF OF PROPOSITION 2

Following theorem 1, to compute the lower bound for the network described in proposition 2, we need to find the first layer for which its total receptive field is greater than $H/2$, and then estimate its total stride and its

$\alpha$-minimal total receptive field, for $\alpha = \lfloor H/2 \rfloor$. In the following claims we analyze the above properties of the given network:

**Claim 5.** *The total stride and total receptive field of the l-th $B \times B$ layer in the given network, i.e. the $(2l-1)$-th GC layer after the representation layer, are given by the following equations:*

$$T_S^{(2l-1)}(S^{(1)}, \ldots, S^{(2l-1)}) = 2^{l-1}$$
$$T_R^{(2l-1)}(R^{(1)}, S^{(1)}, \ldots, R^{(2l-1)}, S^{(2l-1)}) = (2B-1)2^{l-1} - B + 1$$

*Proof.* From eq. 1 it immediately follows that $T_S^{(2l-1)}(S^{(1)}, \ldots, S^{(2l-1)}) = 2^{l-1}$. From eq. 2, the $2 \times 2$ stride 2 layers do not contribute to the receptive field as $R^{(2l)} - S^{(2l)} = 0$, which results in the following equation:

$$T_R^{(2l-1)}(R^{(1)}, S^{(1)}, \ldots, R^{(2l-1)}, S^{(2l-1)}) = B \cdot 2^{l-1} + \sum_{i=1}^{l-1}(B-1)2^{i-1}$$
$$= B \cdot 2^{l-1} + (B-1)(2^{l-1} - 1)$$
$$= (2B-1)2^{l-1} - B + 1$$

$\square$

**Claim 6.** *The $\alpha$-minimal total receptive field for the l-th $B \times B$ layer in the given network, for $\alpha \in \mathbb{N}$ and $2^{l-1} \leq \alpha < 2^l - 1$, always equals $(\alpha + 1)$.*

*Proof.* From eq. 3, the following holds:

$$T_R^{(2l-1,\alpha)} = \underset{\substack{\forall i \in [l], 1 \leq t_{2i-1} \leq B \\ T_R^{(2l-1)}(t_1,1,2,2,t_3,1,\ldots,t_{2l-1},1) > \alpha}}{\mathrm{argmin}} T_R^{(2l-1)}(t_1, 1, 2, 2, t_3, 1, \ldots, t_{2l-1}, 1)$$

$$= \underset{\substack{\forall i \in [l], 1 \leq t_{2i-1} \leq B \\ t_{2l-1} \cdot 2^{l-1} + \sum_{i=1}^{l-1}(t_{2i-1}-1)2^{i-1} > \alpha}}{\mathrm{argmin}} t_{2l-1} \cdot 2^{l-1} + \sum_{i=1}^{l-1}(t_{2i-1} - 1)2^{i-1}$$

Notice that the right term in the equation resembles a binary representation. If we limit $t_{2i-1}$ to the set $\{1, 2\}$, this term can represent any number in the set $\{0, \ldots, 2^{l-1} - 1\}$, and by choosing $t_{2l-1} = 1$, the complete term can represent any number in the set $\{2^{l-1}, \ldots, 2^l - 1\}$, and specifically, for $2^{l-1} \leq \alpha < 2^l - 1$, there exists an assignment for $t_{2i-1} \in \{1, 2\}$ for $i \in [l-1]$ such that this terms equal $(\alpha+1)$, and thus $T_R^{(2l-1,\alpha)} = \alpha+1$. $\square$

With the above general properties for the given network, we can simplify the expression for the lower bound given in theorem 1:

**Claim 7.** *If the l-th $B \times B$ layer in the given network satisfies $T_R^{(2l-1)}(R^{(1)}, S^{(1)}, \ldots, R^{(2l-1)}, S^{(2l-1)}) > H/2$, then the lower bound given in theorem 1 equals to $M^{2^{2L-2l+1}}$*

*Proof.* From the description of the network and the previous claims it holds that $D = M$, $H = 2^L$, $T_S^{(2l-1)} = 2^{l-1}$, and $T_R^{(2l-1,\lfloor H/2 \rfloor)} = 2^{l-1} + 1$. Substituting all the above in eq. 4 results in:

$$(\text{Eq. 4}) = M^{\left\lfloor \frac{2^L - 2^{L-1} - 1}{2^{l-1}} + 1 \right\rfloor \cdot \left\lceil \frac{2^L}{2^{l-1}} \right\rceil}$$
$$= M^{\left\lfloor 2^{L-l}+1-\frac{1}{2^{l-1}} \right\rfloor \cdot 2^{L-l+1}}$$
$$= M^{2^{2L-2l+1}}$$

$\square$

With all of the above claims in place, we our ready to prove proposition 2:

*Proof.* (**of proposition 2**) From claim 5, we can infer which is the first $B \times B$ layer such that its receptive field is greater than $H/2$:

$$(2B - 1) \cdot 2^{l-1} - B + 1 > 2^{L-1}$$

$$\Rightarrow l > \log_2 \frac{2^L + 2B - 2}{2B - 1}$$

$$\Rightarrow l = 1 + \left\lfloor \log_2 \frac{2^L + 2B - 2}{2B - 1} \right\rfloor$$

Combining the above with claim 7, results in:

$$M^{2^{2L-2l+1}} = M^{2^{2L-1-2\left\lfloor \log_2 \frac{2^L+2B-2}{2B-1} \right\rfloor}}$$

$$\geq M^{2^{2L-1-2\log_2 \frac{2^L+2B-2}{2B-1}}}$$

$$(2^L \equiv H) \Rightarrow = M^{\frac{H^2}{2}\left(1+\frac{H-1}{2B-1}\right)^{-2}}$$

$$= M^{\frac{(2B-1)^2}{2} \cdot \left(1+\frac{2B-2}{H}\right)^{-2}}$$

The limits and the special case for $B \leq \frac{H}{5} + 1$ are both direct corollaries of the above expressions. $\qquad\square$

## C.5    PROOF OF THEOREM 2

We begin by proving an analogue of claim 3, where we show that for any given matricization of the grid tensor $\mathcal{A}(\mathbf{h})$, induced by the overlapping network realizing the function $\mathbf{h}$, the matricization rank is exponential. The motivation behind the construction, for when parameters are "unshared", is that we can utilized the fact that there are separate sets of kernels, with local receptive fields the size of the input, for each spatial location. Thus, each kernel can "connect" the index (of the grid tensor) matching its spatial location, with almost any other index, and specifically such that the two indices come from different sets of the matricization $I \uplus J$ of $\mathcal{A}(\mathbf{h})$. For the "shared" case, we simply use polynomially more output channels to simulate the "unshared" case.

**Claim 8.** *For an arbitrary even partition $(I, J)$ of $\{(1,1), \ldots, (H, H)\}$, such that $|I| = |J| = \frac{H^2}{2}$, there exists an assignment to the parameters of the network given in theorem 2, for either the "unshared" or "shared" settings, such that* $\mathrm{rank}\left(\llbracket \mathcal{A}(\mathbf{h}) \rrbracket\right) = M^{\frac{H^2}{2}}$.

*Proof.* Let $(I, J)$ be an arbitrary even partition of $\{(1,1), \ldots, (H, H)\}$, such that $|I| = |J| = \frac{H^2}{2}$, $I = \{i_1, \ldots, i_{|I|}\}$, and $J = \{j_1, \ldots, j_{|J|}\}$, where for $k \in [\frac{H^2}{2}]$ it holds that $i_k < i_{k+1}$ and $j_k < j_{k+1}$ (using lexical ordering), and we assume w.l.o.g. that $i_1 = (1,1)$. We define the set $\{(q_k, p_k)\}_{k=1}^{\frac{H^2}{2}}$ such that $q_k = i_k$ and $p_k = j_k$ if $i_k < j_k$, and otherwise $q_k = j_k$ and $p_k = i_k$.

We prove the t"unshared" case first, where the parameters of the first GC layers are given by $\{(\mathbf{w}^{(c,u,v)}, \mathbf{b}^{(c,u,v)})\}_{c=1,u=1,v=1}^{D,H,H}$, for which we choose the following assignment:

$$w_{m,j,i}^{(c,u,v)} = \begin{cases} (F^{-1})_{m,c} & c \leq M \text{ and } \exists k \in \left[\frac{H^2}{2}\right], q_k = (u,v) \text{ and } (j,i) \in \{(1,1), p_k - q_k + (1,1)\} \\ 0 & \text{Otherwise} \end{cases}$$

$$b_{j,i}^{(c,u,v)} = \begin{cases} 0 & c \leq M \text{ and } \exists k \in \left[\frac{H^2}{2}\right], q_k = (u,v) \text{ and } (j,i) \in \{(1,1), p_k - q_k + (1,1)\} \\ 1_{[c \leq M]} & \text{Otherwise} \end{cases}$$

where for $u, v \in [H]$ such that $q_k = (u, v)$ it holds that $p_k - q_k + (1,1) \in \{(1,1), \ldots, (H, H)\}$ because $q_k \leq p_k$. Similar to the proof of claim 3, we set all layers following the first GC layer such that the following equality holds:

$$(\mathcal{A}(\mathbf{h}))_{d_{(1,1)}, \ldots, d_{(H,H)}} = \prod_{u,s=1}^{H} \sum_{c=1}^{D} \prod_{j,i=1}^{H} \left( b_{ji}^{(c,u,v)} + \sum_{m=1}^{M} w_{mji}^{(c,u,v)} O_{m,u+j-1,v+i-1} \right)$$

Which under the assignment to parameters we chose earlier, it results in:

$$(\mathcal{A}(\mathbf{h}))_{d_{(1,1)},\ldots,d_{(H,H)}} = \prod_{k=1}^{H^2/2} \sum_{c=1}^{D} \left( \sum_{m=1}^{M} (F^{-1})_{m,c} F_{d_{q_k},m} \right) \left( \sum_{m=1}^{M} (F^{-1})_{m,c} F_{d_{p_k},m} \right)$$

$$= \prod_{k=1}^{H^2/2} \sum_{c=1}^{D} (F \cdot F^{-1})_{d_{q_k},c} \cdot (F \cdot F^{-1})_{d_{p_k},c}$$

$$= \prod_{k=1}^{H^2/2} \sum_{c=1}^{D} 1_{[d_{q_k}=c]} \cdot 1_{[d_{p_k}=c]}$$

$$= \prod_{k=1}^{H^2/2} 1_{[d_{q_k}=d_{p_k}]}$$

which means $[\![\mathcal{A}(\mathbf{h})]\!]_{(I,J)}$ equals to the Kronecker product of $\frac{H^2}{2}$ $M \times M$-identity matrices, up to permutations of its rows and columns which do not affect its matrix rank. Thus, $\operatorname{rank}([\![\mathcal{A}(\mathbf{h})]\!]) = M^{\frac{H^2}{2}}$.

For the "shared" setting, we denote the parameters of the first GC layer by $\{(\mathbf{w}^{(d)}, \mathbf{b}^{(d)})\}_{c=1}^{D}$, and set them as:

$$w_{m,j,i}^{(d)} = \begin{cases} (F^{-1})_{m,c} & \left(\exists c \in [M] \exists u, v \in [H], d = cH^2 + uH + v\right) \\ & \text{and } \left(\exists k \in \left[\frac{H^2}{2}\right], q_k = (u,v) \text{ and } (j,i) \in \{(1,1), p_k - q_k + (1,1)\}\right) \\ 0 & \text{Otherwise} \end{cases}$$

$$b_{j,i}^{(d)} = \begin{cases} 0 & \left(\exists c \in [M] \exists u, v \in [H], d = cH^2 + uH + v\right) \\ & \text{and } \left(\exists k \in \left[\frac{H^2}{2}\right], q_k = (u,v) \text{ and } (j,i) \in \{(1,1), p_k - q_k + (1,1)\}\right) \\ 1_{[d \leq MH^2]} & \text{Otherwise} \end{cases}$$

the parameters of the other layers are set as in the "unshared" case, and the proof follows similarly. □

In the preceding claim, we have found a separate example for each matricization, such that the matricization rank is exponential. In the following proof of the theorem 2, we leverage basic properties from measure theory to show that almost everywhere the induced grid tensor has an exponential matricization rank, under every possible even matricization – without explicitly constructing such an example.

*Proof.* (**of theorem 2**) For any even partition $(I, J)$ of $\{(1,1), \ldots, (H, H)\}$, according to claim 8 there exist parameters for which $\operatorname{rank}([\![\mathcal{A}(\mathbf{h})]\!])_{(I,J)} = M^{\frac{H^2}{2}}$, and thus according to lemma 3 the set of parameters for which $\operatorname{rank}([\![\mathcal{A}(\mathbf{h})]\!]) < M^{\frac{H^2}{2}}$ is of measure zero. Since the finite union of sets of measure zero is also of measure zero, then almost everywhere (with respect to the Lebesgue measure) the parameters results in networks such that for all even partitions $\operatorname{rank}([\![\mathcal{A}(\mathbf{h})]\!])_{(I,J)} = M^{\frac{H^2}{2}}$. □

