# OpenReview forum: "On the Expressive Power of Overlapping Architectures of Deep Learning"
_ICLR.cc/2018/Conference — Accept (Poster)_

### Official Review · AnonReviewer2 · 2017-11-28
**It's shown that a small amount of overlap could result in a large increase in expressibility in CNNs. The caveats are that the analysis is not for regular CNNs, and does not say much about approximation.**

**Rating:** 6
**Confidence:** 4

**Review:**

The paper studies the expressive power provided by "overlap" in convolution layers of DNNs.  Instead of ReLU networks with average/max pooling (as is standard in practice), the authors consider linear activations with product pooling.  Such networks, which have been known as convolutional arithmetic circuits, are easier to analyze (due to their connection to tensor decomposition), and provide insight into standard DNNs.

For these networks, the authors show that overlap results in the overall function having a significantly higher rank (exponentially larger) than a function obtained from a network with non-overlapping convolutions (where the stride >= filter width).  The key part of the proof is showing a lower bound on the rank for networks with overlap.  They do so by an argument well-known in this space: showing a lower bound for some particular tensor, and then inferring the bound for a "generic" tensor.

The results are interesting overall, but the paper has many caveats:
1.  the results are only for ConvACs, which are arguably quite different from ReLU networks (the non-linearity in successive non-pooling layers could be important).
2.  it's not clear if the importance of overlap is too surprising (or is a pressing question to understand, as in the case of depth).
3.  the rank of the tensor being high does not preclude approximation (to a very good accuracy) by tensors of much smaller rank.

That said, the results could be of interest to those thinking about minimizing the number of connections in ConvNets, as it gives some intuition about how much overlap might 'suffice'.

I recommend weak accept.

---

> ### Author Response · Authors · 2017-12-04
> **The importance of overlaps and some clarifications of our results**
>
> We thank the reviewer for reading our paper and taking the time reviewing it. Our response follows:
>
> 1. While ConvACs do not model all aspects of ReLU networks (as you noted), they do model what we believe are their most important properties: locality, sharing, and pooling, i.e. the architecture of the network. Additionally, It has already been shown in the past [1] that results on ConvACs could be transfer to ReLU networks with max pooling, which is a topic we wish to address in consecutive works.
> 2. One of the contributions of our paper is to specifically highlight this area that has so far being mostly overlooked compared to the many studies (both empirical and theoretical) that have been focused just on the depth property. Extending our understanding beyond the depth is critical if wish to understand why modern networks work better than the ones that came before, and how we can design better networks in the future. More specifically for the case of overlaps, in many respects the inclusion of overlapping convolutional filters in almost all neural networks used in practice is taken today for granted, while so far there weren’t any theoretical arguments to why that should be the case — in fact, there are actually some recent theoretical arguments to why there shouldn’t be any overlaps, e.g. universality [2], better convergence [3], and simply requiring less computations. On the other hand, practitioners have slowly moved to networks where the overlapping degree is considerably decreased compared to the past (where large convolutional kernels were very commons, while today a mixture of 3x3 and 1x1 convolutions prevail). We explain both phenomena by proving that overlaps lead to exponential expressive efficiency, and that even with the degree of overlaps used in practice this exponential separation is already achieved.
> 3. It is important to emphasize that though our analysis relies on grid tensors, our bounds are on the rank of different matricizations of these tensors, and hence our results could translate to approximation bounds for specific cases by examining the singular values of the matricized grid tensors. Particularly, the example we construct for our lower bound is very close to the identity matrix, and thus non-overlapping networks would not be able to approximate such a grid tensor to a good degree.
>
> References:
> [1] Cohen et al. “Convolutional Rectifier Networks as Generalized Tensor Decompositions” (ICML 2016)
> [2] Cohen et al. “On the Expressive Power of Deep Learning: A Tensor Analysis” (COLT 2016)
> [3] Brutzkus et al. “Globally Optimal Gradient Descent for a ConvNet with Gaussian Inputs” (ICML 2017)

---

### Official Review · AnonReviewer1 · 2017-11-29
**On the Expressive Power of Overlapping Architectures of Deep Learning**

**Rating:** 8
**Confidence:** 3

**Review:**

The paper analyzes the expressivity of convolutional arithmetic circuits (ConvACs), where neighboring neurons in a single layer have overlapping receptive fields. To compare the expressivity of overlapping networks with non-overlapping networks, the paper employs grid tensors computed from the output of the ConvACs.  The grid tensors are matricized and the ranks of the resultant matrices are compared. The paper obtains a lower bound on the rank of the resultant grid tensors, and uses them to show that an exponentially large number of non-overlapping ConvACs are required to approximate the grid tensor of an overlapping ConvACs. Assuming that the result carries over to ConvNets, I find this result to be very interesting.  While overlapped convolutional layers are almost universally used, there has been very little theoretical justification for the same. This paper shows that overlapped ConvACs are exponentially more powerful than their non-overlapping counterparts.

---

> ### Author Response · Authors · 2017-12-10
> **Response to reviewer 1**
>
> We thank the reviewer for taking the time to review our paper and supporting it.

---

### Official Review · AnonReviewer3 · 2017-12-04
**Good idea but some issues.**

**Rating:** 6
**Confidence:** 4

**Review:**

The paper studies convolutional neural networks where the stride is smaller than the convolutional filter size; the so called overlapping convolutional architectures. The main object of study is to quantify the benefits of overlap in convolutional architectures.

The main claim of the paper is Theorem 1, which is that overlapping convolutional architectures are efficient with respect to non-overlapping architectures, i.e., there exists functions in the overlapping architecture which require an exponential increase in size to be represented in the non-overlapping architecture; whereas overlapping architecture can capture within a linear size the functions represented by the non-overlapping architectures. The main workhorse behind the paper is the notion of rank of matricized grid tensors following a paper of Cohen and Shashua which capture the relationship between the inputs and the outputs, the function implemented by the neural network.

(1) The results of the paper hold only for product pooling and linear activation function except for the representation layer, which allows general functions. It is unclear why the generalized convolutional networks are stated with such generality when the results apply only to this special case. That this is the case should be made clear in the title and abstract. The paper makes a point that generalized tensor decompositions can be potentially applied to solve the more general case, but since it is left as future work, the paper should make it clear throughout.

(2) The experiment is minimal and even the given experiment is not described well. What data augmentation was used for the CIFAR-10 dataset? It is only mentioned that the data is augmented with translations and horizontal flips. What is the factor of augmentation? How much translation? These are important because there maybe a much simpler explanation to the benefit of overlap: it is able to detect these translated patterns easily. Indeed, this simple intuition seems to be why the authors chose to make the problem by introducing translations and flips.

(3) It is unclear if the paper resolves the mystery that they set out to solve, which is a reconciliation of the following two observations (a) why are non-overlapping architectures so common? (b) why only slight overlap is used in practice?  The paper seems to claim that since overlapping architectures have higher expressivity that answers (a). It appears that the paper does not answer (b) well: it points out that since there is exponential increase, there is no reason to increase it beyond a particular point. It seems the right resolution will be to show that after the overlap is set to a certain small value, there will be *only* linear increase with increasing overlap; i.e., the paper should show that small overlap networks are efficient with respect to *large* overlap networks; a comparison that does not seem to be made in the paper.

(4) Small typo: the dimensions seem to be wrong in the line below the equation in page 3.

The paper makes important progress on a highly relevant problem using a new methodology (borrowed from a previous paper). However, the writing is hurried and the high-level conclusions are not fully supported by theory and experiments.

---

> ### Author Response · Authors · 2017-12-15
> **Response to Reviewer 3 and a new revision of our submission**
>
> We thank the reviewer for reading our paper and taking the time reviewing it. Our response follows:
>
> 1. We present the concept of Generalized Convolutional Networks to link our overlapping extension of Convolutional Arithmetic Circuits (ConvACs) to standard ConvNets with overlaps — there is more than one way to extend ConvACs to overlapping architecture, and the GC framework we introduced shows why our specific extension is natural with respect to standard ConvNets. Given the reviewer’s suggestion, we have updated our abstract to emphasize that we examine the expressive efficiency of overlapping architectures by theoretically analyzing ConvACs as surrogates to standard ConvNets, while demonstrating empirically that our predictions hold for standard ConvNets as well.
>
> 2. Regarding our experiments:
>
>     * We have added the requested details on or experiments to a new revision of our submission. Specifically, we uniformly sampled translations with a maximal translation of 3 pixels in each direction, i.e. 10% of the dimensions of the CIFAR10 images. We also plan to release the source code for reproducing our results once the double blind phase of ICLR is over.
>
>     * To resolve the concerns of the reviewer regarding an alternative explanation to our experiments that used translation augmentations, we have conducted an additional set of experiments. Instead of relying on spatial transformations, this time we use color augmentations, i.e. we randomly sample a constant shift to the hue, saturation and luminance of each pixel, as well as a randomly sample multiplication factor just for the saturation and luminance (see new revision for details). This augmentation method is based on the one suggested by [1].  These new results exactly follow the behavior we have seen with spatial augmentations: (i) there is a very large gap between the non-overlapping and overlapping case, even when comparing against non-overlapping networks with many more channels, and (ii) when plotted with respect to number of parameters, all overlapping architecture fall on the same curve. Thus, our results cannot be explained simply by some spatial-invariance type argument. Our full results are presented in the revision of our latest revision to our submission.
>
> 3. In the paper we have proved a strong theoretical argument to address observation (a), i.e. why are non-overlapping architectures so common, and a weak theoretical argument to address observation (b), i.e. why only slight overlap is used in practice, that is then augmented by our experiments. Indeed the optimal theoretical argument for observation (b) would be to show an upper on the rank of the grid tensor with respect to the overlapping degree, however, at this point we were only able to show upper bounds for non-overlapping architectures — proving general upper bounds is left for future works. Nevertheless, given our proven lower bounds, we demonstrated (Proposition 2) that even architectures with very little overlapping are already separated from the non-overlapping case. This partially addresses observation (b) by showing that separation from the non-overlapping case does not mean large local receptive fields. Additionally, in our experiments, we see that when we only modify the local receptive fields, then the expressiveness of all overlapping architectures fall on exactly the same curve with respect to the number of parameters, which strengthen our argument that small receptive fields are sufficient, while leaving it as an open conjecture for future theoretical works to prove completely.
>
> 4. We have fixed the typo in our new revision - thank you for pointing it out.
>
> References:
> [1] Dosovitskiy et al. “Discriminative Unsupervised Feature Learning with Convolutional Neural Networks” (NIPS 2014)

---

### Decision · Program_Chairs · 2018-01-29
**ICLR 2018 Conference Acceptance Decision**

**Decision:**

Accept (Poster)

**Comment:**

The paper received scores of 8 (R1), 6 (R2), 6 (R3). R1's review is brief, and also is optimistic that these results demonstrated on ConvACs generalize to real convnets.  R2 and R3 feel this might be a potential problem. R2 advocates weak accept and given that R1 is keen on the paper, the AC feels it can be accepted.